# Unified Multimodal Chain-of-Thought Reward Model through Reinforcement Fine-Tuning

**Yibin Wang**[1,2,4,‡,*] **Zhimin Li**[4,*] **Yuhang Zang**[3,*] **Chunyu Wang**[4],
**Qinglin Lu**[4,†] **Cheng Jin**[1,2] **Jiaqi Wang**[2,3,†]

[1]College of Computer Science and Artificial Intelligence, Fudan University,
[2]Shanghai Innovation Institute [3]Shanghai AI Lab, [4]Hunyuan, Tencent
`https://codegoat24.github.io/UnifiedReward/think`

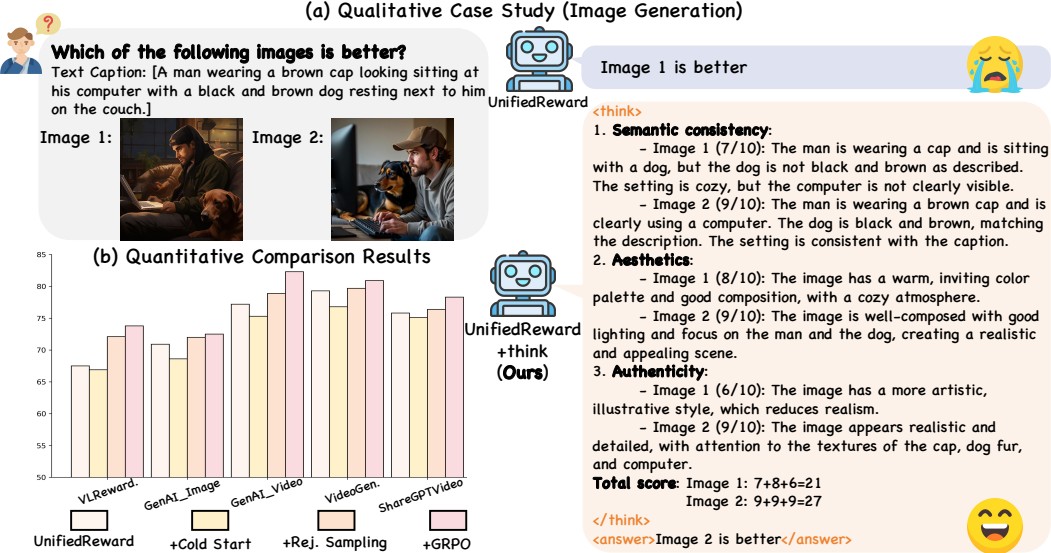

Figure 1: **Overview of Comparison Results.** (a) Our method enables multi-dimensional long CoT reasoning to improve reward signal accuracy. (b) Extensive quantitative results demonstrate our superiority in both vision understanding and generation reward tasks.

## Abstract

Recent advances in multimodal Reward Models (RMs) have shown significant promise in delivering reward signals to align vision models with human preferences. However, current RMs are generally restricted to providing direct responses or engaging in shallow reasoning processes with limited depth, often leading to inaccurate reward signals. We posit that incorporating explicit long chains of thought (CoT) into the reward reasoning process can significantly strengthen their reliability and robustness. Furthermore, we believe that once RMs internalize CoT reasoning, their direct response accuracy can also be improved through implicit reasoning capabilities. To this end, this paper proposes UNIFIEDREWARD-THINK, the first unified multimodal CoT-based reward model, capable of multi-dimensional, step-by-step long-chain reasoning for both visual understanding and generation reward tasks. Specifically, we adopt an exploration-driven reinforcement fine-tuning approach to elicit and incentivize the model's latent complex reasoning

---

*Equal contribution. ‡Project lead. †Corresponding authors.

39th Conference on Neural Information Processing Systems (NeurIPS 2025).

ability: (1) We first use a small amount of image generation preference data to distill the reasoning process of GPT-4o, which is then used for the model's cold start to learn the format and structure of CoT reasoning. (2) Subsequently, by leveraging the model's prior knowledge and generalization capabilities, we prepare large-scale unified multimodal preference data to elicit the model's reasoning process across various vision tasks. During this phase, correct reasoning outputs are retained for rejection sampling to refine the model (3) while incorrect predicted samples are finally used for Group Relative Policy Optimization (GRPO) based reinforcement fine-tuning, enabling the model to explore diverse reasoning paths and optimize for correct and robust solutions. Extensive experiments confirm that incorporating long CoT reasoning significantly enhances the accuracy of reward signals. Notably, after mastering CoT reasoning, the model exhibits implicit reasoning capabilities, allowing it to surpass existing baselines even without explicit reasoning traces.

# 1 Introduction

In recent years, multimodal reward models (RMs) [Wang et al., 2024, 2025d, Zang et al., 2025, Xiong et al., 2024, He et al., 2024, Xu et al., 2024, Liu et al., 2025a, Li et al., 2025, Wang et al., 2025c] have excelled at aligning vision model outputs with human preferences, providing crucial reward signals to guide model training [Wang et al., 2024, 2025d, Liu et al., 2025a, Ouyang et al., 2022, Rafailov et al., 2023, Schulman et al., 2017] and inference [Gulcehre et al., 2023, Snell et al., 2024]. Traditional reward models are typically trained on large-scale human-annotated preference data through supervised fine-tuning (SFT). At test time, most methods [He et al., 2024, Liu et al., 2025a, Xu et al., 2024] directly assign scores or provide pairwise ranking for vision model outputs, relying on the knowledge and intuitions acquired from the training data. While effective, these methods tend to lack interpretability, which makes it difficult for users to understand the underlying reasoning process behind the assigned scores or rankings. To this end, recent studies [Wang et al., 2025d, Xiong et al., 2024, Wang et al., 2024, Li et al., 2025] leverage the generative capabilities of Visual-Language Models (VLMs), enabling RMs to provide concise justifications alongside the assigned reward signals. Despite their success, their reasoning processes often lack rigorous logical structure and the capacity for multi-dimensional, deep analysis, which may result in inaccurate reward signals in complex scenarios or misguided conclusions arising from flawed reasoning processes.

In light of these issues, we posit two key hypotheses: (1) Incorporating explicit long Chains-of-Thought (CoT) into the reward reasoning process is essential for significantly enhancing RM's reliability and robustness; (2) Once the model internalizes this ability, the accuracy of its directly provided reward signals, even without CoT reasoning traces, can also be improved by leveraging its implicit logical thinking capabilities. However, equipping RMs with CoT reasoning using traditional training schemes like SFT poses a highly challenge due to the scarcity of large-scale CoT reward data, as manual annotation requires substantial human resources and time.

In this work, we argue that this challenge can be effectively addressed, as VLMs inherently possess prior knowledge of complex reasoning; what is needed is an effective strategy to elicit and incentivize this capability [Guo et al., 2025]. Therefore, this paper proposes UNIFIEDREWARD-THINK, the first unified multimodal CoT-based reward model, capable of performing multi-dimensional, step-by-step long-chain reasoning across both visual understanding and generation reward tasks. The core idea is to activate the model's latent long-chain reasoning capabilities through limited CoT reward data and to progressively reinforce and refine this capability through exploration-driven reinforcement fine-tuning that optimizes for accurate and robust reasoning patterns. Specifically, our training pipeline, as shown in Fig. 2, consists of three stages: (1) **Cold Start**. We first use a small amount of labeled image generation preference data to distill the reasoning process from GPT-4o [Hurst et al., 2024], filtering for reasoning traces whose final answers align with the ground-truth (GT) labels. These samples serve to cold-start the model training, enabling it to learn the structure and format of long CoT reasoning. (2) **Rejection Sampling**. Next, we prepare large-scale unified multimodal labeled preference data to incentivize the model's CoT reasoning outputs across various visual reward tasks. Reasoning trajectories whose final answers match the GT label are retained for rejection sampling, reinforcing the distribution of correct reasoning patterns. (3) **Group Relative Policy Optimization** (GRPO). Finally, incorrectly reasoned samples are leveraged for GRPO-based reinforcement fine-tuning, enabling the model to explore diverse reasoning paths and optimize toward desirable outcomes defined by our

verified rewards (*i.e.,* format and accuracy rewards). Unlike SFT, which merely imitates predefined answers, GRPO promotes trial-and-error learning by evaluating and refining the model's reasoning outputs based on verified rewards, thus encouraging deeper reasoning discovery rather than passive memorization. Notably, we also propose a **multidimensional CoT reward scoring** strategy to address a widely observed issue, *i.e.,* inconsistency between reasoning and final decisions, where reasoning may appear implausible despite a correct prediction. Specifically, as shown in Fig. 1, by scoring each candidate across various task-relevant dimensions and aggregating the scores from each dimension into a final decision, our approach enforces alignment between reasoning and outcomes, enhancing both the accuracy and reliability of the reward signal. This design allows us to rely solely on the final answer when filtering data during cold-start and rejection sampling, and applying reward supervision in GRPO, while implicitly ensuring the correctness of the reasoning process.

Extensive experiments demonstrate that incorporating long CoT reasoning significantly improves the accuracy and reliability of reward signals. Remarkably, experimental results also prove that once the model internalizes the CoT reasoning ability, it also exhibits strong implicit reasoning capabilities: even when providing direct reward outputs without explicit reasoning traces, it consistently outperforms existing baselines across all vision reward tasks.

**Contributions**: (1) We propose the first unified multimodal CoT reward model, UNIFIEDREWARD-THINK, capable of multi-dimensional, step-by-step long-chain reasoning across both visual understanding and generation tasks; (2) We demonstrate that explicit long CoT reasoning substantially enhances reward model reliability, and once mastered, also strengthens implicit reasoning, enabling more accurate reward signals even without explicit reasoning traces; (3) Extensive experiments validate the superiority of our method compared with existing baselines across all vision reward tasks. We hope our work can unlock reward models' reasoning potential to enhance interpretability, generalization, and alignment, enabling more trustworthy and human-aligned reward signals for multimodal generation and understanding.

## 2 Related work

### 2.1 Multimodal reward models

Multimodal reward models have become increasingly important for aligning vision understanding and generation models with human preferences. A dominant approach is to fine-tune visual-language models (VLMs) [Li et al., 2024a, Bai et al., 2025b], exploiting their powerful multimodal alignment capabilities to learn human judgment-based reward functions. Earlier studies have explored reward models on visual generation [Wang et al., 2024, Liu et al., 2025a, He et al., 2024] and understanding [Zang et al., 2025, Xiong et al., 2024] tasks. For instance, [Wang et al., 2024] collects human feedback and constructs human-rated video datasets to train the reward model, LiFT-Critic, which measures how well the generated videos align with human expectations. [Zang et al., 2025] develops an effective pipeline for constructing multimodal preference datasets and leverages existing high-quality data to train the reward model, IXC-2.5-Reward, enabling accurate evaluation of visual understanding outputs. However, these reward models are task-specific, limiting their adaptability across diverse visual understanding and generation tasks. Therefore, [Wang et al., 2025d] introduces UnifiedReward, a unified reward model capable of assessing both image and video generation and understanding tasks, demonstrating that joint learning across diverse visual tasks can yield substantial mutual benefits.

Despite their effectiveness, these reward models are largely limited to providing direct responses [Liu et al., 2025a, Xu et al., 2024, He et al., 2024, Xu et al., 2023] or engaging in shallow reasoning with limited depth [Wang et al., 2024, 2025d, Xiong et al., 2024] , often resulting in inaccurate or unreliable reward signals in complex scenarios or misguided conclusions arising from flawed reasoning processes. To this end, we propose UNIFIEDREWARD-THINK, the first unified multimodal long CoT-based reward model, enabling multi-dimensional long-chain reasoning for both visual understanding and generation tasks. The core idea is to use reinforcement learning to activate and enhance VLMs' latent reasoning capabilities, which will be discussed in the following section.

### 2.2 Reinforcement learning

Recently, reinforcement learning (RL) techniques have been extensively used to enhance the reasoning capabilities of Large-Language Models (LLMs), enabling them to effectively solve complex problems

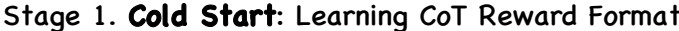

## Stage 1. **Cold Start**: Learning CoT Reward Format

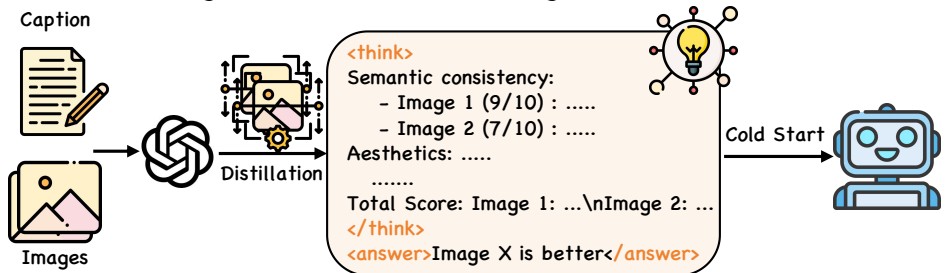

## Stage 2. **Rejection Sampling**: Unified CoT Reward Generalization Fine-tuning

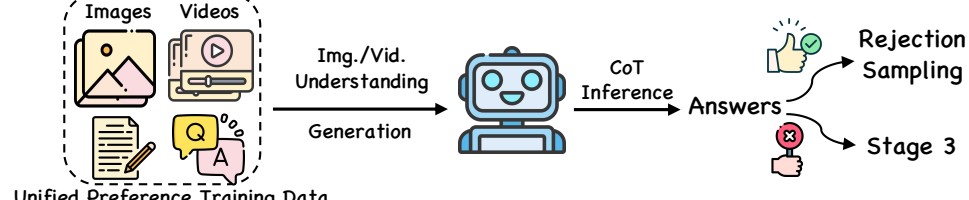

## Stage 3. **GRPO**: Unified CoT Reward Reinforcement Fine-tuning

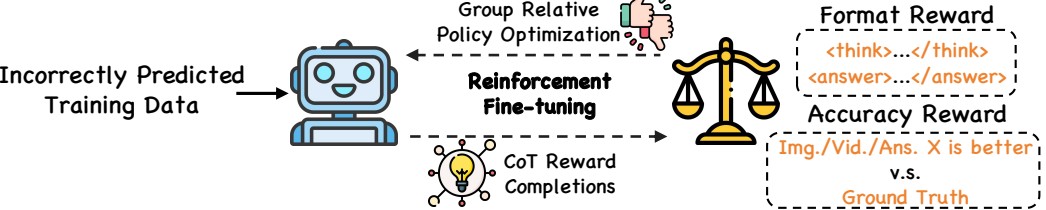

Figure 2: **Method Overview.** (1) **Cold Start**: We first distill GPT-4o's reasoning process on a small amount of labeled image generation preference data to initialize the model's CoT reasoning format; (2) **Rejection Sampling**: Then, we leverage the model's generalization capabilities on large-scale unified multimodal preference data to elicit its CoT reasoning process across various vision tasks. Correctly predicted samples whose final answers match the ground-truth are retained for rejection sampling to further refine the model. (3) **GRPO**: Finally, incorrectly predicted samples are utilized for GRPO-based reinforcement fine-tuning to further enhance step-by-step reasoning capabilities.

[Jaech et al., 2024, Luong et al., 2024, Shao et al., 2024, Yang et al., 2024, Ying et al., 2024, Hui et al., 2024, Jiao et al., 2024, Zhang et al., 2024a,c, Wang et al., 2025a,b]. A significant advancement in this field is Deepseek-R1-Zero [Guo et al., 2025], which introduced a novel approach for developing robust reasoning capabilities using Group Relative Policy Optimization (GRPO). By leveraging rule-based rewards, it enhances reasoning without the need for supervised fine-tuning (SFT). For VLMs, RL has mainly been applied to tasks such as mitigating hallucinations [Sun et al., 2023, Yu et al., 2024a], aligning models with human preferences [Yu et al., 2024b, Zhou et al., 2024], improving visual perception [Liu et al., 2025b], and quality assessment [Li et al., 2025].

However, the application of RL in multimodal reward models is still under exploration. To address this gap, our work introduces a reinforcement fine-tuning strategy that leverages verifiable rewards with GRPO-based RL to integrate long CoT reasoning, enhancing the accuracy of reward signals by enabling multi-dimensional and step-by-step reasoning processes.

## 3 Method

### 3.1 Overview

This work aims to incorporate long Chain-of-Thought (CoT) reasoning into the reward model's decision-making process to enhance the reliability and robustness of reward signals. However, achieving this with traditional training methods like Supervised Fine-Tuning (SFT) remains highly challenging due to the scarcity of large-scale CoT-based reward data. This work posits that Visual-

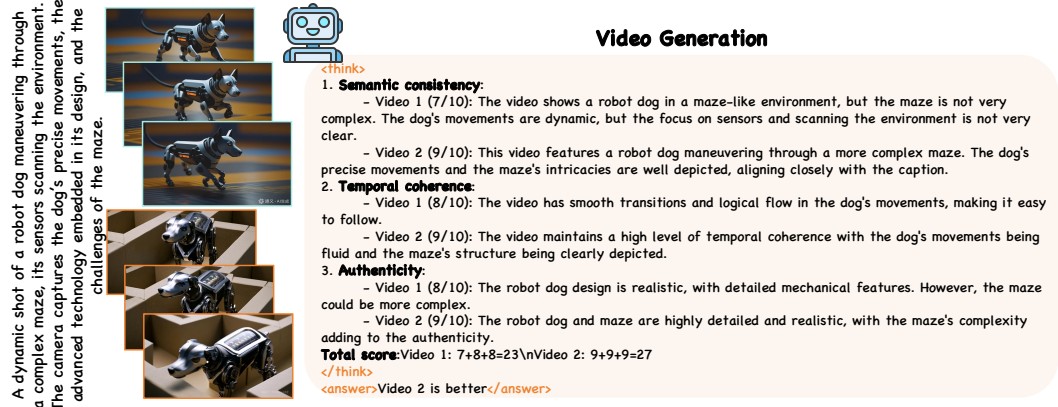

Figure 3: **Qualitative Results of Video Generation CoT Reward Reasoning.** Given a pair of videos and the corresponding caption, our model performs quality assessment across semantic consistency, temporal coherence, and authenticity through CoT reasoning.

Language Models (VLMs) inherently possess prior knowledge of complex reasoning; the key challenge lies in devising an effective strategy to elicit and incentivize this capability. Therefore, we take the first step to propose the unified multimodal CoT-based reward model, UNIFIEDREWARD-THINK, adopting exploration-driven reinforcement fine-tuning to activate and refine the VLM's multi-dimensional and step-by-step long chain reasoning across various vision reward tasks.

Specifically, as shown in Fig. 2, our pipeline includes three key stages: (1) Cold Start: use a small amount of distilled CoT reward data to initialize the reward model with the format and structure of multi-step reasoning (Sec. 3.2); (2) Rejection Sampling: utilize large-scale unified preference data to elicit the model's generalizable CoT reasoning across diverse vision tasks; correctly reasoned samples are retained for rejection sampling to reinforce accurate reasoning patterns (Sec. 3.3); (3) Group Relative Policy Optimization (GRPO): leverage incorrectly reasoned samples for GRPO-based reinforcement fine-tuning to further improve the model's CoT reasoning capabilities (Sec. 3.4). To further ensure consistency between the reasoning process and final decisions, we design an effective multidimensional CoT reward score strategy (Sec. 3.5).

## 3.2 Cold start: learning CoT reward format

In this work, we hypothesize that VLMs inherently possess the potential for complex, long-chain reasoning. However, due to the absence of exposure to reward modeling during pre-training, they often lack a suitable format to articulate such reasoning in this context, which leads to inconsistent or shallow outputs. To address this, we initiate the training with a cold start phase, where a small amount of high-quality CoT reward data distilled from GPT-4o [Hurst et al., 2024] is used to explicitly teach the model the desired reasoning format and structure. Specifically, we begin by preparing a small set of image generation preference data, each consisting of an image pair, a prompt (*i.e.,* instruction and image caption), and the ground-truth (GT) label. These samples are then fed into GPT-4o to generate the detailed long-chain reasoning process and the corresponding final answer. Among the generated samples, we retain only those reasoning trajectories whose final answers align with the GT label, denoted as $(\boldsymbol{x}, \boldsymbol{y})$ where $\boldsymbol{x}$ is the original input, and $\boldsymbol{y} = \{\boldsymbol{y}_1, \boldsymbol{y}_2, \ldots, \boldsymbol{y}_T\}$ is the distilled output, which are subsequently used to cold-start the training. The objective function is defined as:

$$\mathcal{L}_{\text{cold\_start}}(\theta) = -\sum_{i=1}^{T} \log p\left(\boldsymbol{y}_i \mid \boldsymbol{x}, \boldsymbol{y}_{<i}; \theta\right),  \tag{1}$$

where $\theta$ represents the parameters of the reward model. This stage serves to initialize the model's ability to follow a structured CoT reasoning format.

## 3.3 Rejection sampling: unified CoT reward generalization fine-tuning

After learning the CoT reasoning format through cold-start training, we further elicit and expand the model's reasoning capabilities to vision understanding and generation tasks. This is inspired by a

prior study [Wang et al., 2025d], which highlights the benefits of multi-task joint training: reward reasoning abilities learned in one task can generalize well and effectively enhance performance in other vision tasks. Therefore, we apply rejection sampling to reinforce the learning of correct reasoning patterns to improve both reliability and accuracy in vision understanding and generation tasks. Specifically, we first prompt the model to perform generalization CoT reasoning on large-scale unified preference datasets, leveraging its prior knowledge across various tasks (e.g., evaluating video temporal consistency and vision-question-answer accuracy). In vision generation tasks, each input sample consists of a textual prompt (instruction and caption) and a corresponding image/video pair, whereas in vision understanding tasks, it comprises a textual prompt (instruction and query) along with an image/video. Then, we retain only the reasoning samples whose trajectories lead to a final answer consistent with the ground-truth label, while discarding incorrect reasoning. The training objective for this stage is similar to Eqn. 1, but it utilizes the filtered reasoning data of diverse vision tasks $(x', y')$ obtained through rejection sampling.

This process concentrates the training distribution around accurate reasoning patterns and enhances the model's generalization ability across diverse visual domains.

### 3.4 GRPO: unified CoT reward reinforcement fine-tuning

In the rejection sampling stage, a subset of challenging data featuring intricate reasoning patterns is filtered, which the model has yet to fully comprehend and master. To ensure the model fully learns the underlying knowledge in the training dataset and further enhances its reasoning ability, we introduce GRPO-based reinforcement fine-tuning [Guo et al., 2025]. Specifically, in GRPO, the policy model $\pi_\theta$ generates multiple candidate responses for a given input, which are evaluated using predefined verifiable reward functions, providing corresponding reward signals. These signals guide policy updates, encouraging alignment with high-quality reasoning while constraining deviations from the reference model $\pi_{\text{ref}}$. This approach enables the model to explore diverse reasoning processes, guiding it toward the correct reasoning trajectory and improving its ability to handle complex scenarios. In the following, we will first introduce the design of verifiable rewards in our work, followed by a detailed description of the GRPO training process.

#### 3.4.1 Verifiable reward

In GRPO, verifiable rewards are essential for guiding the model's learning by offering rule-based feedback. In this work, we employ two types of verifiable rewards *i.e.,* format reward and accuracy reward, to ensure both the quality and accuracy of the model's responses:

**Format reward** ensures that the generated response follows a specific reasoning structure, which is critical for maintaining clarity and consistency in the reasoning process. In this case, the response is expected to contain two key tags: $\langle$think$\rangle$ and $\langle$answer$\rangle$. These tags are used to delineate the model's reasoning process and the final answer, respectively. If both tags are present and properly formatted in the response, the reward $R_{fmt}$ is set to 1. Otherwise, the reward $R_{fmt}$ is 0. This mechanism helps reinforce the importance of structuring the response. Although most VLMs already exhibit strong instruction-following capabilities, the format reward can still serve as a safety check to filter out occasional formatting errors, which, though minor, may negatively affect model performance.

**Accuracy reward** evaluates whether the final answer output $o$ (*i.e.,* Image/Video/Answer $X$ is better), enclosed within the $\langle$answer$\rangle$ tag, exactly matches the ground truth. This reward serves as a reliable signal to ensure that the model produces correct answers. If $o$ matches the ground truth precisely, the reward is set to 1; otherwise, it is 0. Formally, the accuracy reward is defined as:

$$R_{acc} = \begin{cases} 1 & \text{if } o = \text{ground truth} \\ 0 & \text{otherwise.} \end{cases}$$

This is crucial for reinforcing accurate reasoning and encouraging the model to generate correct responses. Finally, the overall verifiable reward $R$ is formulated as:

$$R = R_{fmt} + R_{acc}.$$

By incorporating both the format and accuracy rewards into the GRPO training, we provide the model with explicit feedback that encourages it to generate responses that are both well-structured and factually accurate.

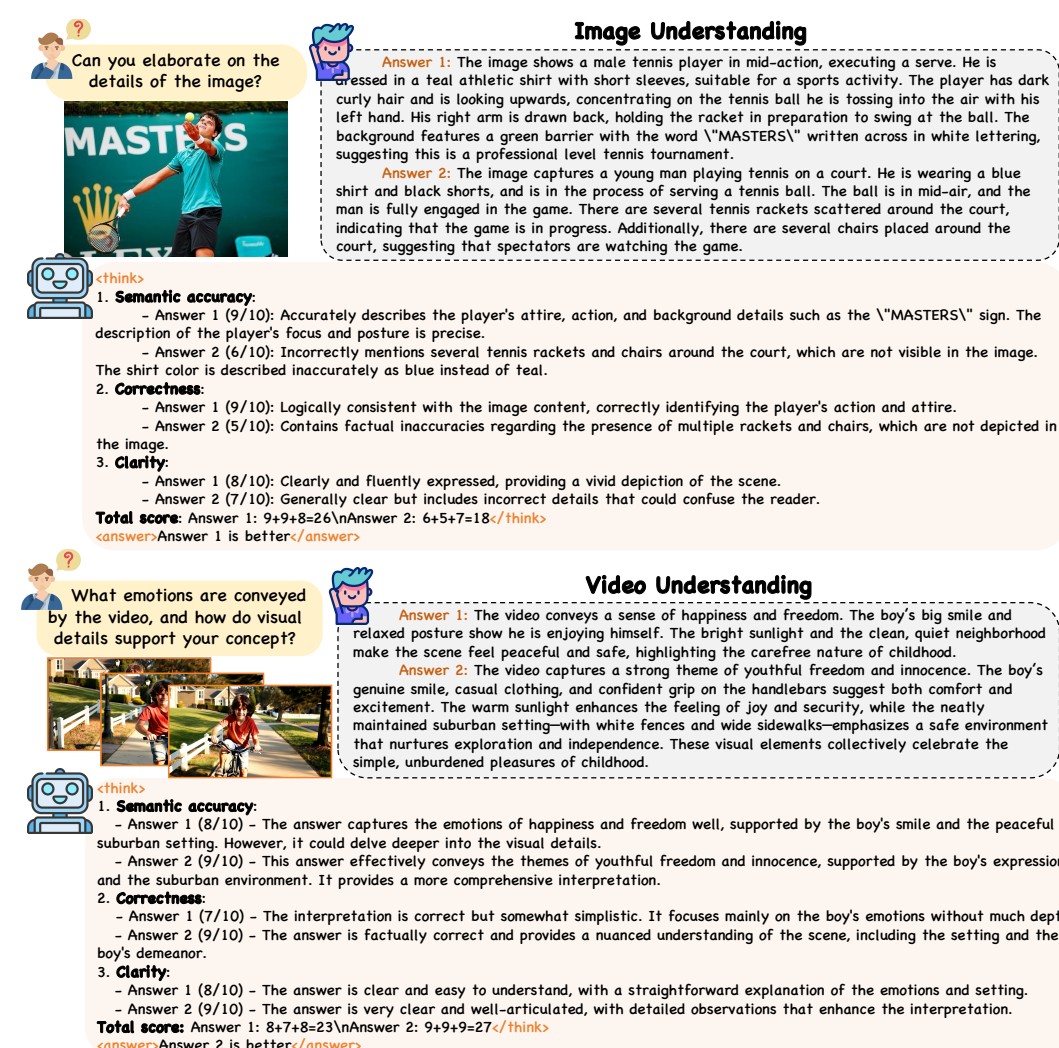

Figure 4: **Qualitative Cases of Image and Video Understanding CoT Reward Reasoning.** Given an image or video, a query, and a pair of candidate answers, our model performs quality assessment across semantic accuracy, factual correctness, and clarity through CoT reasoning.

### 3.4.2 Reinforcement fine-tuning

Given an input $\boldsymbol{x}$, GRPO first samples $N$ distinct responses $\{o^{(1)}, o^{(2)}, \ldots, o^{(N)}\}$ from the previous policy model $\pi_{\theta_{\mathrm{old}}}$. Each response is evaluated using our verifiable rewards, *i.e.,* format and accuracy rewards, resulting in corresponding reward scores $\{R^{(1)}, R^{(2)}, \ldots, R^{(N)}\}$. Then, GRPO normalizes these scores and quantifies the relative quality of each response by computing the advantage of each response using the standardized reward:

$$\hat{A}^{(i)} = \frac{R^{(i)} - \mathrm{mean}(\{R^{(1)}, \ldots, R^{(N)}\})}{\mathrm{std}(\{R^{(1)}, \ldots, R^{(N)}\})},$$

where $\hat{A}^{(i)}$ quantifies the relative quality of the $i$-th response in comparison to other candidates within the same sampled group.

Then, GRPO estimates magnitude of the policy update by computing the likelihood ratio of each response under the new policy $\pi_{\theta_{\mathrm{new}}}$ relative to the old policy $\pi_{\theta_{\mathrm{old}}}$, defined as:

$$r^{(i)} = \frac{\pi_{\theta_{\mathrm{new}}}(o^{(i)} \mid \boldsymbol{x})}{\pi_{\theta_{\mathrm{old}}}(o^{(i)} \mid \boldsymbol{x})}.$$

To stabilize training and avoid overly aggressive updates, the ratio is clipped to a bounded interval $[1 - \delta, 1 + \delta]$. Moreover, to ensure that the updated policy does not diverge significantly from a fixed reference model $\pi_{\text{ref}}$, a KL divergence penalty term is introduced with a weighting factor $\beta$. The final optimization objective is defined as:

$$L_{grpo}(\theta) = \mathbb{E}_{\boldsymbol{x} \sim \mathcal{X}, o^{(i)} \sim \pi_{\theta_{\text{old}}}} \left[ \min \left( r^{(i)} \hat{A}^{(i)}, \text{clip}(r^{(i)}, 1 - \delta, 1 + \delta) \hat{A}^{(i)} \right) - \beta \cdot D_{\text{KL}}(\pi_{\theta_{\text{new}}} \parallel \pi_{\text{ref}}) \right],$$

where $\mathcal{X}$ denotes the set of training sample input and $D_{\text{KL}}(\cdot \parallel \cdot)$ denotes the KL divergence.

By integrating normalized reward advantages, clipped importance sampling, and reference-model regularization, GRPO enables stable and effective policy optimization, driving our model toward generating higher-quality, verifiably correct CoT reasoning paths.

### 3.5 Multidimensional CoT reward scoring strategy

In the cold-start and rejection sampling stages, we filter data by retaining only those samples whose final answers align with the ground truth, and in the GRPO stage, supervision is likewise applied solely to the final answers. However, both GPT-4o and existing reward models often suffer from inconsistencies between reasoning traces and final decisions, where reasoning may appear implausible even if the prediction is correct, leading to unreliable supervision. To address this issue, we propose a multidimensional CoT reward scoring strategy that tightly couples the reasoning process with the final score. Specifically, the model evaluates each candidate (image, video, or textual response) along multiple task-relevant dimensions, aggregates the dimension-wise scores into a final decision, and thereby enforces consistency between reasoning and outcomes (Figs. 1, 3, 4). By grounding the final prediction in multidimensional and interpretable reasoning processes, our design implicitly ensures reasoning correctness even when data filtering or supervision relies solely on the final answer, as in all training stages, thereby enhancing both the accuracy and reliability of our reward model.

## 4 Experiments

### 4.1 Experimental setup

**Datasets.** For *Image Generation*, we utilize HPD (25.6K) [Christodoulou and Kuhlmann-Jørgensen, 2024], OIP (7.4K)[1], EvalMuse (3K) [Han et al., 2024], all preprocessed by [Wang et al., 2025d], as well as OpenAI-4o_t2i_human_preference (6.7K) collected by Rapidata[2]. For *Video Generation*, we employ VideoDPO (10K) [Liu et al., 2024] and Text2Video-Human Preferences (5.7K), also collected by Rapidata. For *Image Understanding*, we sample 30K data from LLaVA-Critic-113K [Xiong et al., 2024]. For *Video Understanding*, we adopt ShareGPTVideo-DPO (17K) [Zhang et al., 2024b]. In the cold-start stage, we distill image generation CoT reward reasoning samples from GPT-4o, constructing our ImageGen-CoT-Reward-5K. The input data are randomly sampled from the image generation datasets, with the remaining data reserved for the subsequent training stages.

**Reward model.** We adopt UnifiedReward [Wang et al., 2025d] as our base model, which is capable of assessing both image/video generation and understanding. We leverage its strong performance and extensive prior knowledge in visual perception and generation, and further activate its latent capacity for long-chain CoT reasoning.

**Training details.** For both the cold-start and rejection sampling stages, training is performed with a batch size of 1, 16 gradient accumulation steps, a learning rate of $2.5 \times 10^{-6}$, and a warm-up ratio of 0.3, using 8 NVIDIA H100 (80GB) GPUs. For GRPO, training is conducted with a batch size of 1, a single gradient accumulation step, a learning rate of $1 \times 10^{-6}$, and a KL penalty coefficient of $\beta = 0.04$. The number of generated responses $N$ is set to 8, using 64 NVIDIA H20 (97GB) GPUs.

**Evaluations.** We evaluate image and video understanding reward assessment on VLRewardBench [Li et al., 2024b] and ShareGPTVideo [Zhang et al., 2024b], using 5K test samples, respectively. For generation evaluation, we adopt GenAI-Bench [Jiang et al., 2024], which covers both image and video reward benchmarks. Additionally, we utilize VideoGen-RewardBench [Liu et al., 2025a] to further assess video generation.

---

[1]https://huggingface.co/datasets/data-is-better-together/open-image-preferences-v1-binarized
[2]https://huggingface.co/datasets/Rapidata

Table 1: **Image Understanding Assessment Comparison.** We evaluate baselines across different understanding aspects on VLRewardBench.

| Models | General | Hallu. | Reason. | Overall Accuracy | Macro Accuracy |
|---|---|---|---|---|---|
| Gemini-1.5-Pro | 50.8 | 72.5 | 64.2 | 67.2 | 62.5 |
| GPT-4o | 49.1 | 67.6 | **70.5** | 65.8 | 62.4 |
| LLaVA-Critic | 47.4 | 38.5 | 53.8 | 46.9 | 46.6 |
| UnifiedReward | 76.5 | 58.1 | 65.1 | 67.5 | 66.6 |
| Ours (w/o CoT) | 77.9 | 70.5 | 65.4 | 73.1 | 71.3 |
| **Ours** | **78.1** | **72.7** | 66.0 | **73.8** | **72.3** |

Table 2: **Image and Video Generation Assessment Comparison.** "tau" indicates that accuracy is calculated with ties, and "diff" excludes tied pairs when calculating accuracy.

| Method | Image Generation | | Method | Video Generation | | | |
|---|---|---|---|---|---|---|---|
| | GenAI-Bench | | | GenAI-Bench | | VideoGen-Reward | |
| | tau | diff | | tau | diff | tau | diff |
| PickScore | 53.2 | 67.2 | VideoScore | 46.2 | 70.6 | 42.1 | 49.9 |
| HPSv2 | 51.6 | 68.4 | LiFT | 41.2 | 60.1 | 40.6 | 58.3 |
| ImageReward | 47.8 | 65.0 | VisionReward | 52.1 | 73.1 | 57.4 | 68.2 |
| VisionReward | 46.8 | 66.4 | VideoReward | 50.2 | 73.3 | 60.1 | 73.9 |
| UnifiedReward | **54.8** | 70.9 | UnifiedReward | **60.7** | 77.2 | **66.6** | 79.3 |
| Ours (w/o CoT) | 54.1 | 71.9 | Ours (w/o CoT) | 57.8 | 81.6 | 64.9 | 79.9 |
| **Ours** | - | **72.5** | **Ours** | - | **82.3** | - | **80.5** |

## 4.2 Comparison results

The quantitative results, presented in Tabs. 1, 2, and Fig. 1, demonstrate the superiority of our model across visual generation and understanding tasks. In image and video generation tasks, our model does not account for tie scenarios during evaluation, since such cases were not included in the training data. Nevertheless, it consistently outperforms existing methods across all other evaluations. Notably, compared to the base model UnifiedReward, incorporating multi-dimensional and multi-step reasoning yields substantial performance gains across all tasks. In particular, our model achieves significant improvements in image understanding reward tasks. This is intuitive, as CoT reasoning in vision tasks fundamentally centers on a deeper understanding of the visual content. Additionally, we explore whether our model, after learning long CoT reasoning, can improve the accuracy of direct reward signals, *i.e.,* without explicit CoT reasoning. Experimental results show that the model still outperforms existing methods by leveraging implicit logical reasoning, with only a slight drop in performance compared to explicit CoT reasoning. These results strongly validate the effectiveness and superiority of our approach. Our qualitative results are provided in Figs. 1, 3, and 4.

## 4.3 Ablation studies

**Ablation of each training stage.** We conduct ablation studies to assess the effectiveness of each training stage. As shown in Tabs. 3 and 4, after the cold start phase, though the model learns the CoT reasoning format, it still struggles with accurate reward prediction. Notably, introducing rejection sampling leads to clear improvements by retaining correctly reasoned samples for supervised fine-tuning, thereby reinforcing desirable reasoning patterns. Further, the GRPO stage yields the most substantial gains, as it focuses on previously mispredicted cases, allowing the model to explore multiple reasoning paths and converge on more accurate solutions. These results highlight the complementary roles of each stage and demonstrate how our staged training strategy progressively enhances CoT-based reward modeling.

**Ablation of GRPO without CoT reasoning.** To further validate the necessity of learning the CoT reasoning process, we evaluate a GRPO variant that removes CoT and directly optimizes reward predictions based on final answers. As shown in Tabs. 3 and 4, although this approach yields slight improvements over the baseline, the gains are significantly limited. This suggests that learning

Table 3: **Ablation Results of Image Understanding Assessment.** We conduct ablation experiments under different settings and evaluate them across multiple aspects on VLRewardBench.

| Models | General | Hallu. | Reason. | Overall Accuracy | Macro Accuracy |
|---|---|---|---|---|---|
| UnifiedReward | 76.5 | 58.1 | 65.1 | 67.5 | 66.6 |
| +GRPO (w/o CoT) | 77.8 | 59.1 | 65.5 | 69.0 | 67.4 |
| UnifiedReward | 76.5 | 58.1 | 65.1 | 67.5 | 66.6 |
| +cold start | 76.0 | 56.7 | 65.2 | 66.9 | 66.0 |
| +rejection sampling | 77.6 | 64.9 | 65.4 | 72.1 | 69.3 |
| **+GRPO(Ours)** | **78.1** | **72.7** | **66.0** | **73.8** | **72.3** |

Table 4: **Ablation Results of Image and Video Generation Assessment.** "tau" indicates that accuracy is calculated with ties, and "diff" excludes tied pairs when calculating accuracy.

| Method | Image Generation | | Method | Video Generation | | | |
|---|---|---|---|---|---|---|---|
| | GenAI-Bench | | | GenAI-Bench | | VideoGen-Reward | |
| | tau | diff | | tau | diff | tau | diff |
| UnifiedReward | 54.8 | 70.9 | UnifiedReward | 60.7 | 77.2 | 66.6 | 79.3 |
| +GRPO (w/o CoT) | 54.1 | 71.3 | +GRPO (w/o CoT) | 56.7 | 78.4 | 64.8 | 79.5 |
| UnifiedReward | 54.8 | 70.9 | UnifiedReward | 60.7 | 77.2 | 66.6 | 79.3 |
| +cold start | - | 68.6 | +cold start | - | 75.3 | - | 76.8 |
| +rejection sampling | - | 72.0 | +rejection sampling | - | 78.9 | - | 79.7 |
| **+GRPO(Ours)** | - | **72.5** | **+GRPO(Ours)** | - | **82.3** | - | **80.5** |

from final answers alone fails to teach the model the underlying reasoning process. In contrast, our CoT-based GRPO guides the model to explore multiple reasoning trajectories and gradually converge toward the correct path, leading to deeper understanding and more robust generalization. These results show that the effectiveness of our GRPO-based reinforcement fine-tuning stems from explicitly modeling the reasoning process, rather than simply reinforcing the final answer.

## 5 Limitations and future works

While our method introduces long-form CoT reasoning to improve reward modeling, this inevitably increases inference time during reasoning. However, we show that once the model has mastered CoT reasoning, it can leverage implicit reasoning to enhance answer accuracy even without explicitly generating CoT traces. This suggests strong internalization of the reasoning process. Besides, although our reinforcement fine-tuning strategy successfully activates the model's latent long CoT reasoning ability using only a small amount of high-quality data, the prior study [Yue et al., 2025] has shown that reinforcement learning cannot fundamentally extend a model's capability: it can only amplify the potential already acquired during supervised fine-tuning (SFT). Therefore, to further push the boundaries of CoT-based reward reasoning, scaling up high-quality CoT supervision still remains a promising direction.

## 6 Conclusion

We propose UNIFIEDREWARD-THINK, the first unified multimodal CoT reward model capable of multi-dimensional, step-by-step reliable reward reasoning for both visual understanding and generation tasks. Specifically, we adopt an exploration-driven reinforcement fine-tuning approach to elicit and incentivize the model's latent complex reasoning ability, including three key stages: cold start, rejection sampling, and GRPO-based reinforcement fine-tuning. Our experiments demonstrate that CoT reasoning not only improves the accuracy and robustness of reward signals but also equips the model with strong implicit reasoning capabilities, enabling superior performance even without explicit CoT outputs. We hope our work can unlock reward models' reasoning potential to enhance interpretability and generalization, enabling more trustworthy and human-aligned reward signals for multimodal understanding and generation.

# 7 Acknowledgments

This work was supported by the National Natural Science Foundation of China (Grant No. 62176064), AI for Science Foundation of Fudan University (FudanX24AI028), and Shanghai Innovation Institute.

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

Table 5: **Ablation of Rejection Sampling Training Stage.** We conduct an ablation study to assess the effect of the rejection sampling stage on different benchmarks.

| Method | VLReward | GenAI-Image | GenAI-Video | VideoGen |
|---|---|---|---|---|
| Baseline | 67.5 | 70.9 | 77.2 | 79.3 |
| **UnifiedReward-Think** | **73.8** | **72.5** | **82.3** | **80.5** |
| w/o Rejection Sampling | 70.4 | 71.6 | 79.2 | 79.8 |

## A   Further methodological insights

**Cold start only with image generation preference data: why it works.** Our experiments demonstrate that using a small amount of high-quality image generation CoT reward reasoning data, instead of distilling data for every task, is sufficient for the model to learn the CoT reasoning format and structure across all visual reward tasks. The underlying reason lies in the fact that video tasks can be seen as multi-image understanding problems. Video frames, like images, also require recognizing objects, spatial relationships, and context. Once the model masters CoT reasoning for images, it can naturally extend this reasoning to videos by leveraging its inherent prior knowledge of temporal dynamics and sequential visual understanding. Therefore, by learning CoT reasoning from images, the model can seamlessly generalize to both static and dynamic visual tasks, eliminating the need for separate distillation for each task.

**Rejection sampling for unified reward generalization fine-tuning: why we need it.** After the cold start, the model has already internalized the format and structure of CoT reasoning. With prior knowledge across tasks, it can generate accurate CoT-based reward analyses for many simple scenarios. However, directly applying GRPO to the entire training set would be inefficient and computationally costly. Besides, GRPO offers limited gains on samples that the model has already mastered. Therefore, we first apply rejection sampling to filter out cases that the model already performs well on. This not only reduces training cost but also reinforces the distribution of correct reasoning patterns by prioritizing high-confidence outputs. More challenging or ambiguous samples are passed to the GRPO stage, where the model explores diverse reasoning trajectories and gradually learns to prefer more accurate solutions. Ablation results of this training stage are provided in Tab. 5.

**Why we trust the CoT reasoning when the final answer is correct during GRPO.** In MLLMs, a common failure mode during CoT reasoning is the disconnect between the reasoning steps and the final answer: models may produce plausible reasoning chains but ultimately rely on shortcuts or intuition to generate the conclusion. To mitigate this, we explicitly structure the CoT process by having the model score each image across multiple dimensions and then aggregate these scores to derive the final decision. This enforces a step-by-step alignment between intermediate reasoning and outcome, ensuring that a correct final answer emerges from a coherent and interpretable reasoning process. As a result, in GRPO, verifying the correctness of the final answer implicitly validates the reasoning trajectory, offering a principled yet efficient way to supervise complex CoT generation.

**Why learning can be generalized to different tasks.** In this work, our unified multimodal CoT reward model is designed to support both vision generation and understanding tasks across both images and videos. Although the downstream tasks may differ, the input-output structure to the reward model is unified: in all cases, the model receives a combination of visual content (image or video) and caption or textual response, and outputs a reward judgment signal indicating the relative quality or preference between candidate outputs. Because of this consistent structure, the reward model learns to evaluate visual-text pairs in a task-agnostic manner, allowing it to generalize across different modalities and task types. This design enables us to train the reward model jointly on diverse datasets and apply it effectively across vision reward tasks.

## B   Robustness on different baselines

To further demonstrate the robustness of our method across different base models, we additionally train UNIFIEDREWARD-THINK on Qwen2.5-VL-7b [Bai et al., 2025a]. As shown in Tab. 6, leveraging the stronger capability of the base model, the Qwen2.5-VL–based model achieves consistent improvements.

Table 6: **Performance Comparison on Different Backbones.** We compare the performance of UNIFIEDREWARD-THINK trained on LLaVA-OneVision and Qwen2.5-VL.

| UnifiedReward-Think | GenAI-Bench | | VLRewardBench | | | | |
|---|---|---|---|---|---|---|---|
| | Image | Video | General | Hallu. | Reason. | Overall Accuracy | Macro Accuracy |
| LLaVA-OneVision-7B | 72.5 | 82.3 | 78.1 | 72.7 | 66.0 | 73.8 | 72.3 |
| Qwen2.5-VL-7B | **76.6** | **84.0** | **78.3** | **75.4** | **74.2** | **75.9** | **74.6** |

Table 7: **Exploring CoT Distillation from Different Models.** We compare the impact of distilling CoT samples from GPT-4o versus Qwen2.5-VL.

| Method | VLReward | GenAI-Image | GenAI-Video | VideoGen |
|---|---|---|---|---|
| Baseline | 67.5 | 70.9 | 77.2 | 79.3 |
| Ours (distilling GPT-4o) | **73.8** | 72.5 | **82.3** | **80.5** |
| Ours (distilling Qwen2.5-VL-72b) | 73.2 | **73.6** | 81.7 | 80.2 |

## C  Role of distilled CoT samples on cold-start stage

In our pipeline, the cold-start CoT data serves primarily to teach the model the structure and format of CoT-style reasoning, rather than to enhance reward decision quality. The generalization and strengthening of the model's reward discrimination capability are achieved through the subsequent stages of rejection sampling and GRPO, which operate on large-scale unified multimodal vision preference data and iteratively reinforce correct reasoning trajectories. Our experimental results in Tab. 7 also show that even when using distilled CoT data from weaker models (Qwen2.5VL-72b) instead of GPT-4o for cold-start, the model can still achieve comparable performance after subsequent reinforcement through rejection sampling and GRPO.

## D  Details of our cold-start dataset: *ImageGen-CoT-Reward-5K*

In this work, we distill a small set of CoT reward reasoning samples for image generation from GPT-4o [Hurst et al., 2024], which are used to cold-start our model and teach it the format and structure of CoT reasoning. Specifically, we randomly select 10K samples from HPD, EvalMuse, and OIP datasets and input them into GPT-4o for distillation. To ensure consistency between the long-chain reasoning process and final decision, we carefully designed the CoT reasoning format: the model is prompted to independently score each image along multiple evaluation dimensions, and the final judgment is made based on the aggregated scores across all dimensions. This structured approach mitigates inconsistencies between intermediate reasoning and final conclusions. An example prompt template is shown in Fig. 5. We retain only those CoT outputs whose final conclusion matches the ground truth preference label. After filtering, we obtain a total of 5K high-quality CoT reward reasoning samples for image generation, constructing our cold-start dataset: ImageGen-CoT-Reward-5K.

## E  More experimental details

### E.1  Base model.

This work adopts UnifiedReward [Wang et al., 2025d] as our base architecture, which unifies image/video generation and understanding reward tasks, demonstrating the mutual benefits of multi-task learning and achieving strong performance across various vision reward benchmarks. However, its reasoning is limited to direct responses or shallow thinking, lacking the capability for long and structured CoT reasoning, which may lead to lower accuracy and weaker interpretability in complex scenarios. Inspired by this work, we build upon UnifiedReward and further activate its latent long CoT reasoning capabilities across different vision reward tasks, aiming to enhance the accuracy and robustness of the reward signals.

## E.2 Reward model baselines.

We compare our method against a series of strong reward model baselines across both image and video domains, covering generation and understanding tasks.

**PickScore** [Kirstain et al., 2023] is a text-to-image preference model that integrates CLIP-based vision-language features with a reward modeling strategy inspired by InstructGPT. It is trained on the Pick-a-Pic dataset to align image generation outputs with human preferences.

**HPSv2** [Christodoulou and Kuhlmann-Jørgensen, 2024] builds upon CLIP and is fine-tuned using the HPD_v2 [Christodoulou and Kuhlmann-Jørgensen, 2024] to predict human preferences over generated images. It demonstrates strong performance in pairwise ranking tasks and serves as a representative image generation reward baseline.

**ImageReward** [Xu et al., 2023] is trained on a large-scale preference dataset containing 137k human expert comparisons through both rating and ranking. It is specifically designed to capture subtle aspects of human preferences in text-to-image generation quality.

**LLaVA-Critic** [Xiong et al., 2024] extends large language models for evaluating image understanding through a critic-style framework. It is trained on high-quality instruction-following data covering diverse criteria such as accuracy, relevance, and hallucination, supporting both pointwise and pairwise evaluation.

**VisionReward** [Xu et al., 2024] introduces a fine-grained, multi-dimensional evaluation framework for both image and video domains. It trains separate reward models tailored to human preferences collected through carefully curated datasets, offering strong baselines for visual content assessment.

**VideoScore** [He et al., 2024] focuses on assessing video generation quality. It is trained on the VideoFeedback dataset comprising human-annotated scores over 37.6K videos, each evaluated across multiple aspects including fidelity, consistency, and alignment.

**LiFT-Critic** [Wang et al., 2024] is a reward model developed under the LiFT framework, which aligns text-to-video models using human feedback. Trained on LiFT-HRA—a dataset containing over 10K human-labeled samples with both scores and rationales—it captures detailed human evaluation signals across multiple dimensions.

**VideoReward** [Liu et al., 2025a] offers a multi-dimensional assessment for video generation tasks. It is trained on a large-scale 182K dataset of human-labeled comparisons collected from outputs of 12 video generation models, providing strong performance on complex video benchmarks.

**UnifiedReward** Wang et al. [2025d] serves as our base architecture. It leverages multi-task learning across diverse image and video generation and understanding datasets. By unifying multimodal reward tasks into a single framework, UnifiedReward demonstrates mutual enhancement effects and establishes a solid baseline for holistic visual reward modeling.

**Our UnifiedReward-Think** extends UnifiedReward by integrating explicit long CoT reasoning across both visual understanding and generation tasks. Through a three-stage training pipeline—including cold start to learning CoT reward format, rejection sampling for unified CoT reward generalization fine-tuning, and GRPO for unified CoT reward reinforcement fine-tuning, the model achieves stronger accuracy and interpretability in reward assessment. It also generalizes well without explicit reasoning, leveraging implicit CoT capabilities for robust performance.

## E.3 Evaluation benchmarks

**VLRewardBench** [Li et al., 2024b] serves as a diverse benchmark for evaluating image understanding capabilities, featuring 1,250 carefully curated samples across general vision-language queries, hallucination detection, and complex reasoning. To ensure robust evaluation, response orders are randomly shuffled during testing.

**ShareGPTVideo** [Zhang et al., 2024b] provides large-scale video-caption pairs and human preference data, covering various aspects of video understanding such as temporal reasoning, spatial relations, and factual grounding. We 3K for evaluation in our reward modeling experiments.

**GenAI-Bench** [Jiang et al., 2024] is a multimodal generation benchmark designed to assess how well models align with human preferences across image and video generation tasks. We adopt its image and video generation subsets for evaluating generative reward performance.

**VideoGen-RewardBench** [Liu et al., 2025a] offers a large-scale benchmark tailored for evaluating video reward models, consisting of 26.5k video pairs labeled by humans. Each pair is ranked according to multiple criteria, and we use the Overall Quality scores to benchmark the performance of the model.

## F   Regarding performance on "Tau" metric

In both GenAI-Bench and VideoGen benchmarks shown in Tab. 2, the "Tau" metric is specifically designed to evaluate model behavior in *tie* cases, where two images or videos are equally preferred. In this work, our approach is explicitly focused on enhancing the model's discriminative ability, that is, its capacity to distinguish between better and worse samples. While existing reward models are generally effective at identifying outputs with large quality differences, they often struggle to make fine-grained distinctions between two high-quality candidates. To address this gap, we introduce a multidimensional CoT reasoning process, aimed at improving the model's sensitivity to subtle differences in quality. As a result, we intentionally exclude tie cases from our training data, and our method is not optimized for these scenarios. This design choice is deliberate and aligns with practical objectives in vision-based reinforcement learning, where reward models are expected to make clear and decisive judgments between competing outputs in order to effectively guide policy optimization. In contrast, baseline models were specially trained on datasets that include such tie scenarios, enabling them to perform well in such situations. Although baseline models may achieve higher "Tau" scores, our method, both with and without CoT reasoning, consistently demonstrates superior performance in "Diff." metrics where a quality difference exists between the two candidates. We believe these results convincingly demonstrate the effectiveness of our proposed training approach.

## G   Prompting templates and more qualitative cases

We provide more qualitative cases across diverse vision tasks with prompting templates in Figs. 5, 6, 7 and 8.

## H   Societal impacts

Our work introduces a unified multimodal CoT reward model capable of high-quality, interpretable assessment across diverse multimodal tasks. This advancement can significantly enhance the alignment of generative models with human preferences in real-world applications such as AI-assisted content creation, and education. By improving both the accuracy and interpretability of reward signals, our method contributes more transparent and controllable AI behaviors, potentially increasing public trust in generative technologies. However, as reward models become more capable and general, they may also be misused to reinforce harmful biases in generation models, especially if the training data or preference annotations reflect subjective or skewed human values. We encourage future work to further examine the ethical implications of large-scale reward modeling and to include fairness-aware training strategies.

## I   Ethical statement

In this work, we affirm our commitment to ethical research practices and responsible innovation. To the best of our knowledge, this study does not involve any data, methodologies, or applications that raise ethical concerns. All experiments and analyses were conducted in compliance with established ethical guidelines, ensuring the integrity and transparency of our research process.

**Image Generation** 🧑❓

a photo of a boy eating ramen in a reastaurant

Given a caption and two images generated based on this caption, please analyze in detail the two provided images. Evaluate them on various dimensions such as semantic consistency (how closely the image content aligns with the caption), aesthetics (composition, color usage, artistic expression), authenticity (realism and attention to detail), and any other factors you deem relevant. For each evaluation dimension, provide a score between 1-10 for both images (e.g., Image 1: 8/10, Image 2: 6/10) and provide a concise rationale for the score. Calculate the total score for each image by summing all dimension scores. Use a chain-of-thought process to detail your reasoning steps, and enclose all your detailed reasoning within <think> and </think> tags. Then, in the <answer> tag, output exactly one of the following strings: \'Image 1 is better\' or \'Image 2 is better\' based on the total scores. No additional text is allowed in the <answer> section.

Example output format:
<think>
1. Semantic consistency: Image 1 (9/10) – ...; Image 2 (7/10) – ...
2. Aesthetics: Image 2 (8/10) – ...; Image 1 (8/10) – ...
3. Authenticity: Image 1 (8/10) – ...; Image 2 (5/10) – ...
[Additional dimensions if any]: Image 2 (8/10) – ...; Image 1 (6/10) – ...
Total score:
Image 1: 9+8+8+6=31
Image 2: 7+8+5+8=28
</think>
<answer>Image 1 is better</answer>
Note: In the example above, scores and the final answer are placeholders meant only to demonstrate the format. Your actual evaluation should be based on the quality of two given images.

Your task is provided as follows:
Text Caption: [{prompt}]

<think>
**1. Semantic consistency:**
  – Image 1 (9/10): The image shows a boy eating ramen in a restaurant setting, which aligns well with the caption. The presence of a bowl of ramen and the restaurant environment are clear.
  – Image 2 (8/10): This image also depicts a boy eating ramen in a restaurant. However, the focus on the boy's face is more intense, slightly reducing the emphasis on the restaurant setting.
**2. Aesthetics:**
  – Image 1 (8/10): The composition is balanced, with warm lighting that enhances the dining atmosphere. The focus on the boy and the ramen is clear and visually appealing.
  – Image 2 (7/10): The image has a good composition, but the focus on the boy's face might distract from the overall aesthetic. The lighting is slightly harsher, affecting the ambiance.
**3. Authenticity:**
  – Image 1 (9/10): The details in the image, such as the texture of the ramen and the restaurant background, are realistic and convincing.
  – Image 2 (7/10): While the image is realistic, the focus on the boy's face and the lighting make it slightly less authentic in terms of capturing a typical restaurant dining experience.
**Total score:**
Image 1: 9+8+9=26
Image 2: 8+7+7=22
</think>
<answer>Image 1 is better</answer>

Figure 5: **More Qualitative Results of Image Generation CoT Reward Reasoning.** Given a pair of images and the corresponding caption, our model performs quality assessment across semantic consistency, aesthetics, and authenticity through CoT reasoning.

## Video Generation

The camera remains still, a woman with long blonde hair and wearing a green blouse turns on the faucet with her hands, and water flows out from the faucet, the background is a modern kitchen, bright natural lighting.

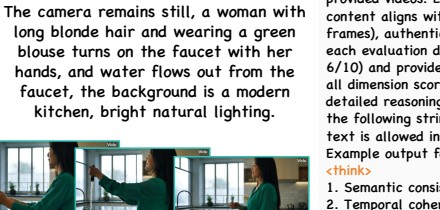

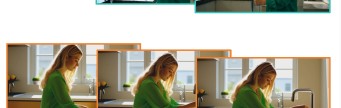
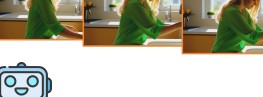
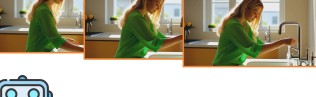

Given a caption and two videos generated based on this caption, please analyze in detail the two provided videos. Evaluate them on various dimensions such as semantic consistency (how closely the video content aligns with the caption), temporal coherence (smoothness and logical flow of motion across frames), authenticity (realism and attention to detail), and any other factors you deem relevant. For each evaluation dimension, provide a score between 1–10 for both videos (e.g., Video 1: 8/10, Video 2: 6/10) and provide a concise rationale for the score. Calculate the total score for each video by summing all dimension scores. Use a chain-of-thought process to detail your reasoning steps, and enclose all your detailed reasoning within <think> and </think> tags. Then, in the <answer> tag, output exactly one of the following strings: 'Video 1 is better' or 'Video 2 is better' based on the total scores. No additional text is allowed in the <answer> section.

Example output format:
<think>
1. Semantic consistency: Video 1 (9/10) – ...; Video 2 (7/10) – ...
2. Temporal coherence: Video 1 (8/10) – ...; Video 2 (6/10) – ...
3. Authenticity: Video 1 (7/10) – ...; Video 2 (5/10) – ...
[Additional dimensions if any]: Video 2 (8/10) – ...; Video 1 (6/10) – ...
Total score:
Video 1: 9+8+7+6=30
Video 2: 7+6+5+8=26
</think>
<answer>Video 1 is better</answer>
Note: In the example above, scores and the final answer are placeholders meant only to demonstrate the format. Your actual evaluation should be based on the quality of two given videos.
Your task is provided as follows:
Text Caption: [{prompt}]

<think>
**1. Semantic consistency:**
   – Video 1 (5/10): The video shows a woman in a green blouse, but her hair is dark, not blonde. The faucet is turned on, but the water flow is not visible. The setting is a modern kitchen, aligning with the caption.
   – Video 2 (9/10): The video features a woman with long blonde hair and a green blouse, matching the caption. The faucet is turned on, and water flows out, consistent with the description.
**2. Temporal coherence:**
   – Video 1 (4/10): The woman's actions are not smooth, and the faucet turning on is not clearly depicted. The scene lacks a coherent flow.
   – Video 2 (9/10): The video shows a clear, smooth action of turning on the faucet with water flowing, maintaining a logical sequence.
**3. Authenticity:**
   – Video 1 (6/10): The setting and attire are realistic, but the lack of visible water flow reduces authenticity.
   – Video 2 (9/10): The video is realistic, with natural lighting and a believable scene of water flowing from the faucet.
**Total score:**
Video 1: 5+4+6=15
Video 2: 9+9+9=27
</think>
<answer>Video 2 is better</answer>

Figure 6: **More Qualitative Results of Video Generation CoT Reward Reasoning.** Given a pair of videos and the corresponding caption, our model performs quality assessment across semantic consistency, temporal coherence, and authenticity through CoT reasoning.

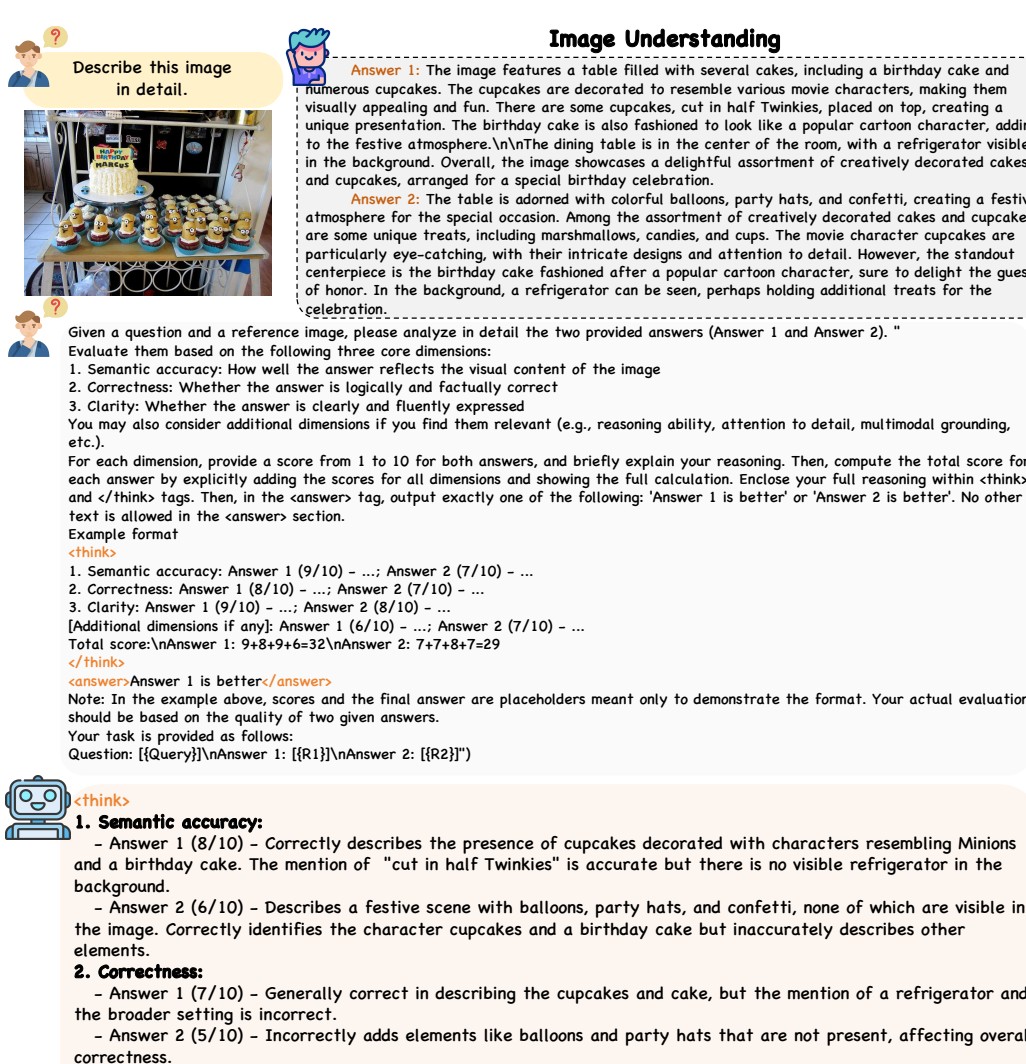

Figure 7: **More Qualitative Cases of Image Understanding CoT Reward Reasoning.** Given an image, a query, and a pair of candidate answers, our model performs quality assessment across semantic accuracy, factual correctness, and clarity through CoT reasoning.

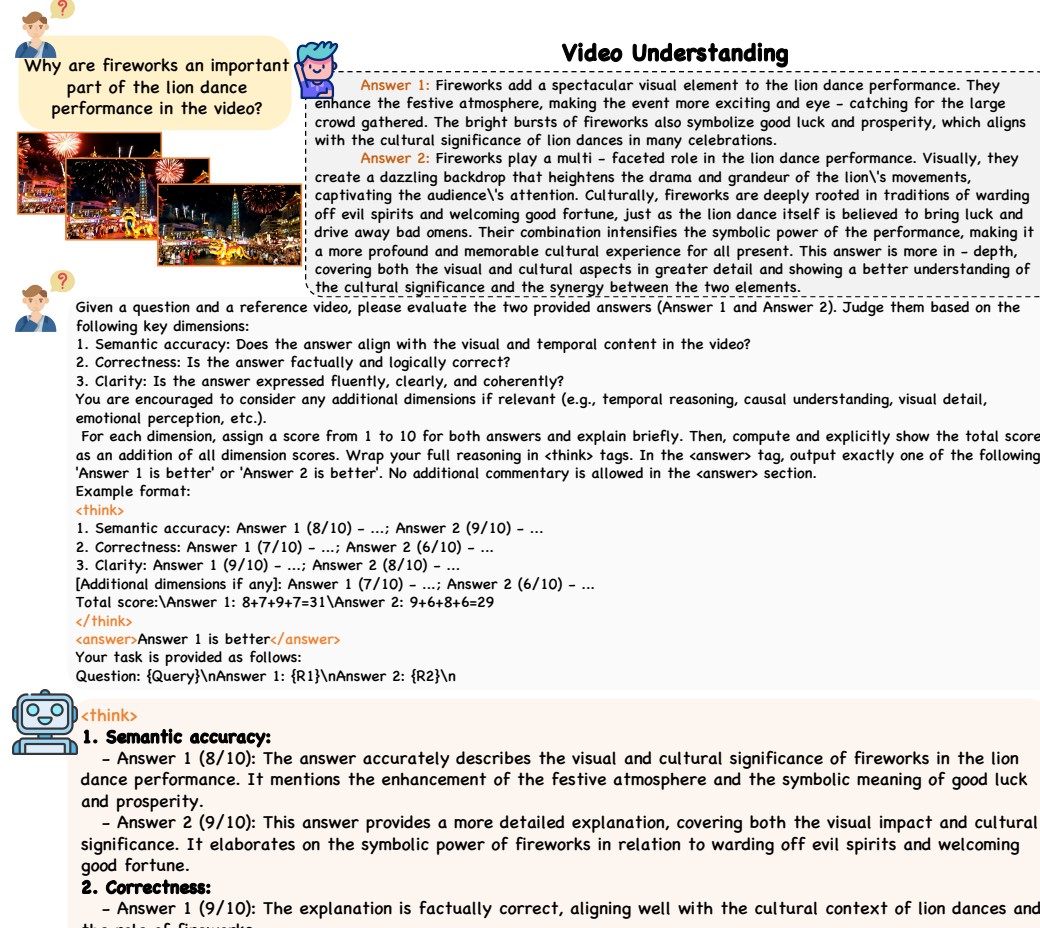

**Video Understanding**

**Why are fireworks an important part of the lion dance performance in the video?**

**Answer 1:** Fireworks add a spectacular visual element to the lion dance performance. They enhance the festive atmosphere, making the event more exciting and eye - catching for the large crowd gathered. The bright bursts of fireworks also symbolize good luck and prosperity, which aligns with the cultural significance of lion dances in many celebrations.

**Answer 2:** Fireworks play a multi - faceted role in the lion dance performance. Visually, they create a dazzling backdrop that heightens the drama and grandeur of the lion\'s movements, captivating the audience\'s attention. Culturally, fireworks are deeply rooted in traditions of warding off evil spirits and welcoming good fortune, just as the lion dance itself is believed to bring luck and drive away bad omens. Their combination intensifies the symbolic power of the performance, making it a more profound and memorable cultural experience for all present. This answer is more in - depth, covering both the visual and cultural aspects in greater detail and showing a better understanding of the cultural significance and the synergy between the two elements.

Given a question and a reference video, please evaluate the two provided answers (Answer 1 and Answer 2). Judge them based on the following key dimensions:
1. Semantic accuracy: Does the answer align with the visual and temporal content in the video?
2. Correctness: Is the answer factually and logically correct?
3. Clarity: Is the answer expressed fluently, clearly, and coherently?
You are encouraged to consider any additional dimensions if relevant (e.g., temporal reasoning, causal understanding, visual detail, emotional perception, etc.).
For each dimension, assign a score from 1 to 10 for both answers and explain briefly. Then, compute and explicitly show the total score as an addition of all dimension scores. Wrap your full reasoning in <think> tags. In the <answer> tag, output exactly one of the following: 'Answer 1 is better' or 'Answer 2 is better'. No additional commentary is allowed in the <answer> section.
Example format:
<think>
1. Semantic accuracy: Answer 1 (8/10) – ...; Answer 2 (9/10) – ...
2. Correctness: Answer 1 (7/10) – ...; Answer 2 (6/10) – ...
3. Clarity: Answer 1 (9/10) – ...; Answer 2 (8/10) – ...
[Additional dimensions if any]: Answer 1 (7/10) – ...; Answer 2 (6/10) – ...
Total score:\Answer 1: 8+7+9+7=31\Answer 2: 9+6+8+6=29
</think>
<answer>Answer 1 is better</answer>
Your task is provided as follows:
Question: {Query}\nAnswer 1: {R1}\nAnswer 2: {R2}\n

<think>
**1. Semantic accuracy:**
   – Answer 1 (8/10): The answer accurately describes the visual and cultural significance of fireworks in the lion dance performance. It mentions the enhancement of the festive atmosphere and the symbolic meaning of good luck and prosperity.
   – Answer 2 (9/10): This answer provides a more detailed explanation, covering both the visual impact and cultural significance. It elaborates on the symbolic power of fireworks in relation to warding off evil spirits and welcoming good fortune.
**2. Correctness:**
   – Answer 1 (9/10): The explanation is factually correct, aligning well with the cultural context of lion dances and the role of fireworks.
   – Answer 2 (10/10): This answer is also factually correct and provides a comprehensive understanding of the cultural and symbolic aspects of fireworks in the context of the lion dance.
**3. Clarity:**
   – Answer 1 (8/10): The answer is clear and well-structured, effectively conveying the importance of fireworks in the performance.
   – Answer 2 (9/10): The answer is very clear and detailed, offering a thorough explanation that enhances understanding.
**Total score:**
Answer 1: 8+9+8=25
Answer 2: 9+10+9=28
</think>
<answer>Answer 2 is better</answer>

Figure 8: **More Qualitative Cases of Video Understanding CoT Reward Reasoning.** Given a video, a query, and a pair of candidate answers, our model performs quality assessment across semantic accuracy, factual correctness, and clarity through CoT reasoning.

