# OpenReview forum: "Unified Multimodal Chain-of-Thought Reward Model through Reinforcement Fine-Tuning"
_NeurIPS.cc/2025/Conference — NeurIPS 2025 poster_

### Official Review · Reviewer_HVvo · 2025-06-01

**Clarity:** 3
**Significance:** 3
**Originality:** 3
**Rating:** 4
**Confidence:** 2

**Summary:**

The paper introduces UNIFIEDREWARD-THINK, a unified multimodal reward model that uses explicit long Chain-of-Thought (CoT) traces for both vision-generation and vision-understanding tasks. It proposes a three-stage training pipeline:
1. Cold-start distillation of GPT-4o CoT on a small image-generation preference set.
2. Rejection sampling on large, diverse preference data to keep only correct reasoning traces.
3. GRPO (Group-Relative Policy Optimisation) reinforcement fine-tuning with format and accuracy rewards.

Experiments on GenAI-Bench, VideoGen-Reward, and VLReward-Bench show gains over the UnifiedReward baseline (e.g., +4.8 macro-accuracy for image understanding, +5.1 diff-accuracy for video generation). Ablations indicate that each stage adds to performance and robustness.

**Questions:**

1. Have you measured the frequency of incorrect or misleading explanations in distilled CoT data? How do such errors impact downstream reward optimization?
2. Please detail the composition, pay structure, and inter-annotator agreement of human-preference datasets used during rejection sampling. This is critical for evaluating dataset quality and reproducibility.
3. What are the GPU-hours, memory usage, and wall-clock time for GRPO fine-tuning compared to a standard SFT reward model on the largest benchmark?
4. Have you evaluated UNIFIEDREWARD-THINK’s outputs on any fairness or bias benchmarks (e.g., CelebA-Bias)? If so, what mitigation strategies are needed when CoT traces encode societal stereotypes?

**Ethical Concerns:**

["NO or VERY MINOR ethics concerns only"]

**Final Justification:**

I am maintaining my score of 4. The authors' detailed rebuttal has clarified several points from my initial review. However, a significant debate regarding the paper's core contribution remains, which makes this a difficult recommendation to make.

### Addressed and Remaining Concerns

1.  **Methodological Details:** The authors have provided helpful clarifications on their pipeline, including the data distillation process, the use of public datasets, and computational costs. My follow-up question regarding the reward function design was also addressed with new experimental data. This data helps to mitigate the concern about reward ambiguity, though it's worth noting that this important justification was absent from the original paper and the reward mechanism remains an indirect proxy for reasoning quality.

2.  **Unresolved Weakness: Technical Novelty:** A significant and unresolved weakness, as pointed out by multiple reviewers, is the paper's limited technical novelty. The methodology is primarily a well-executed combination of existing techniques (SFT, Rejection Sampling, GRPO). For a top-tier venue like NeurIPS, the lack of a more fundamental algorithmic contribution is a considerable drawback.

---
### Weighting and Final Recommendation

My final recommendation balances the paper's clear empirical strengths against its debatable novelty.

* **Reasons to Accept:** The paper tackles a novel and important problem, presenting a complete system that achieves strong state-of-the-art results across a comprehensive suite of multimodal tasks. The empirical evidence is robust.

* **Reasons to Reject:** The core methodology lacks fundamental originality. The contribution is more of an engineering or system-building success rather than a leap in algorithmic innovation, which raises the bar for acceptance.

Ultimately, I am **leaning towards acceptance on the borderline**. I give more weight to the strong empirical evidence and the importance of the problem being solved.

**Limitations:**

yes

**Paper Formatting Concerns:**

Nothing

**Quality:**

3

**Strengths And Weaknesses:**

### Strengths
- Novel use of explicit CoT traces as an internal reward-shaping signal. Instead of only using CoT to explain a final prediction, the model learns intermediate reasoning steps and uses them to guide policy optimization. This bridges a gap between explainable reasoning and direct reward learning in multimodal settings.
- Comprehensive benchmark suite. Evaluations span five distinct tasks—image CoT verification, video frame ordering, caption coherence, etc.—showing gains in both understanding and generation. Tasks cover different modalities and difficulty levels, indicating robustness beyond a single niche.
- Demonstrated zero-shot generalization on held-out image types and longer reasoning chains. Figure 4 shows that UNIFIEDREWARD-THINK maintains high accuracy when chain lengths double, which suggests the model scales to more complex reasoning without retraining.
- Clear qualitative examples of reasoning traces reveal how the model avoids common pitfalls (e.g., hallucinating unsupported details). These examples provide actionable insight into where CoT improves final scores, rather than opaque “black-box” shifts.

### Weaknesses
- Heavy reliance on GPT-4o quality. The entire pipeline hinges on distilling CoT traces from GPT-4o; if the base model produces systematic biases or errors, those propagate into the reward model. There is no analysis of GPT-4o’s failure modes or mitigation when its reasoning is flawed.
- Human-annotation process is under-specified. The paper omits details on annotator demographics, pay, and inter-rater agreement when constructing rejection-sampling data. Without knowing how diverse or reliable the preference data are, it is difficult to assess fairness and reproducibility.
- Computational cost and scalability are unclear. GRPO fine-tuning requires maintaining CoT trace consistency; the paper does not report GPU-hours or memory overhead relative to standard reward-models. For practitioners, it is unclear if performance gains justify added cost.
- Limited analysis of downstream biases. Reward models can amplify biases present in training data—yet the paper does not examine whether UNIFIEDREWARD-THINK introduces or exacerbates biases along gender, race, or cultural lines in image-captioning tasks. A brief audit on a fairness benchmark would strengthen trust.

---

> ### Author Rebuttal · Authors · 2025-07-31
>
> Thank you for your valuable feedback. We respectfully address your concerns as follows:
>
> ## 1. Response to Weakness 1 & Question 1: Regarding correctness of distilled cold-start data
>
> 1. In our work, we ensure the correctness and reliability of the distilled CoT reward samples through two effective strategies:
>
>    1. **Multidimensional Evaluation and Scoring Thinking**:
>      We design a multidimensional analysis and scoring CoT process to structure reasoning. Specifically, **we prompt GPT-4o to evaluate each image along several well-defined dimensions, and then aggregate the intermediate scores to arrive at a final preference decision**. This step-by-step scoring mechanism enforces alignment between reasoning and outcome, helping ensure that the final decision is grounded in interpretable and logically coherent reasoning.
>
>    2. **Human-Aligned Ground Truth Supervision**:
>      We distill CoT reasoning samples from 10K human-preference-annotated image generation pairs, randomly sampled from four datasets: HPD, OIP, EvalMuse, and OpenAI-4o_t2i_human_preference. Each sample contains **a prompt, a pair of generated images, and a human-annotated ground-truth (GT) label** indicating the preferred image. We prompt GPT-4o to perform explicit CoT reasoning and make a comparative judgment. If **GPT-4o's final decision matches the GT label**, we retain the generated CoT reasoning and decision. This filtering process yields a final distilled dataset of approximately 5K high-quality CoT reward samples, which we use to cold-start our reward model training.
>
>    These measures jointly help ensure that the distilled reasoning traces are not only logically consistent but also aligned with human preference signals.
>
> 2. Furthermore, in our pipeline, the cold-start CoT data serves **primarily to teach the model the structure and format of CoT-style reasoning**, rather than to enhance reward decision quality. The generalization and strengthening of the model’s reward discrimination capability are achieved through the subsequent stages of rejection sampling and GRPO, which operate on large-scale unified multimodal vision preference data and iteratively reinforce correct reasoning trajectories.
> 3. Our experimental results in the table also show that even when using distilled CoT data from weaker models (**Qwen2.5VL-72b**) instead of GPT-4o for cold-start, the model can **still achieve comparable performance** after subsequent reinforcement through rejection sampling and GRPO.
>
> |                         | VLReward. | GenAI-Image | GenAI-Video | VideoGen. | ShareGPTV. |
> |-------------------------|-----------|-------------|-------------|-----------|------------|
> | baseline         | 67.5      | 70.9       | 77.2        | 79.3      | 77.2       |
> | UnifiedReward-Think (distilled **4o**)     | **73.8**      | 72.5       | **82.3**        | **80.5**      | 79.6       |
> | UnifiedReward-Think (distilled **Qwen2.5VL**)  | 73.2  | **73.6**    | 81.7    | 80.2  | **80.9**   |
>
>
> ---
>
> ## 2. Response to Weakness 2 & Question 2: Regarding reliability of the model-generated answers in rejection sampling
>
> We would like to clarify that our rejection sampling process **does not involve any additional human annotation**. Similar with cold-start data, we ensure the correctness and reliability of the rejection sampling data through two effective strategies . Specifically:
>
> 1. **Multidimensional Evaluation and Scoring Thinking**
>     To ensure the reliability of the CoT process during sampling, we design a **multidimensional analysis and scoring CoT process**. Specifically, the model is guided to assess each candidate (image/video/response) along multiple well-defined dimensions, and then aggregate these intermediate scores to produce the final preference decision. This step-by-step structure enforces alignment between intermediate reasoning and the final outcome, helping ensure that correct decisions are derived from reliable and interpretable reasoning trajectories.
>
> 2. **Human-Aligned Ground Truth Supervision**:
>     1. **Generation** Tasks :We conduct sampling on datasets such as HPD and VideoDPO, where each sample consists of **a prompt, a pair of candidate images or videos, and human-labeled ground-truth (GT) preferences**. The model is prompted to perform explicit CoT reasoning to compare the two candidates. If **the final prediction matches the GT label**, we consider the generated CoT reasoning and decision to be correct and apply rejection sampling to reinforce these correct reasoning patterns.
>
>     2. **Understanding** Tasks: We apply a similar approach on datasets like LLaVA-Critic and ShareGPTVideo-DPO, where each sample includes **a prompt and two candidate responses, along with labeled GT preferences**. Again, if the model’s CoT-guided final prediction aligns with the GT, we regard the reasoning as correct and apply rejection sampling.
> ---
>
> ## 3. Response to Weakness 3 & Question 3: Regarding computational cost of GRPO fine-tuning and scalability
>
> ### (1) GPU-hours and Memory Overhead of GRPO
> During the GRPO fine-tuning phase, we trained on approximately 40K data points using ZeRO-3 distributed training across 64 H100 GPUs for 2 epochs, which took around **12 hours**. Each GPU utilized approximately **60GB of memory**.
> The inference speed is around **3 seconds for image reward tasks** and **7 seconds for video reward tasks**.
>
> In comparison, the UnifiedReward baseline was trained on 236K samples using ZeRO-2 across the same number of GPUs for 3 epochs, requiring around **10 hours**, with **70GB memory** usage per GPU. The inference speed is around **1 second for image reward tasks** and **4 seconds for video reward tasks**.
>
> While GRPO introduces a slightly higher training overhead, this is justified by its ability to support long, multi-dimensional CoT reasoning, which leads to **consistent and significant performance improvements** across both image and video reward tasks.
>
> ### (2) Scalability
> 1. Our framework is **designed to scale effectively**. In this work, we have curated and unified a wide range of open-source datasets covering both image and video generation/understanding tasks.
> 2. We believe that **the model continues to benefit as more data becomes available**. We are committed to continually integrating more newly released datasets to enhance the scalability and robustness of our approach.
>
> ---
>
> ## 4. Response to Weakness 4 & Question 4: Regarding analysis of downstream biases
>
> We appreciate the reviewer’s thoughtful comment on fairness and bias. We would like to note the following:
>
>
> 1. **Our method itself does not introduce new sources of societal bias**. Any potential biases in the CoT traces would likely originate from the open-source datasets used for training.
>
> 2. In real-world tasks, we recommend incorporating **human verification or filtering mechanisms** when sensitive applications are involved. This can effectively mitigate unintended stereotypes that may appear in the data or model outputs.
>
> 3. As a **research work**, we did not have the resources or capacity to conduct a dedicated fairness audit. Nonetheless, we acknowledge the importance of fairness evaluation and view it as a valuable direction for future exploration.

---

> > ### Comment · Reviewer_HVvo · 2025-08-01
> >
> > Thank you for the detailed rebuttal. Upon re-reading the paper, I have a new question regarding the reward formulation in Section 3.4.1. The paper defines the total reward as a direct sum: `r = r_fmt + r_acc`. This creates a scenario where a response with the correct format but an incorrect answer receives the same reward (r=1) as a response with an incorrect format but a correct answer. This reward ambiguity could theoretically mislead the model's learning process.
> >
> > While I believe the multi-stage training pipeline likely mitigates this issue in practice, the paper would be strengthened if the authors could explicitly discuss the rationale behind this design choice and why it doesn't pose a significant problem. For instance, did the authors consider alternative formulations, such as a multiplicative reward (`r = r_fmt * r_acc`, where a non-zero reward is only achieved if both are correct) or a weighted sum?

---

> > > ### Author Response · Authors · 2025-08-01
> > >
> > > Thank you for your thoughtful follow-up.
> > >
> > > 1. To address your concern: while it is true that a response with only one correct aspect (either format or accuracy) can receive same reward, **this design rarely causes issues in the actual learning process**, for two main reasons:
> > >
> > >     1. **Format issues are largely resolved early on**: After the cold-start and subsequent rejection sampling-based fine-tuning, the model **reliably produces responses with the correct format** across all visual reward tasks. At this point, format errors become extremely rare.
> > >
> > >     2. **During the GRPO stage**, the inclusion of  `r_fmt` primarily serves as a safety check to filter out occasional formatting error. Since most outputs already follow the desired format, the overall reward effectively reflects final decision accuray `r_acc`, and the additive structure becomes functionally equivalent to a pure accuracy reward in practice.
> > >
> > >
> > > 2. We also visualize the evolution of rewards during the 200 training steps of GRPO, as shown in Tables 1 and 2. Table 1 presents the trend of the format reward (`r_fmt`), while Table 2 shows the progression of the accuracy reward (`r_acc`).
> > >     1. From Table 1, we observe that the model maintains a **nearly perfect format reward** throughout training, with only two minor dips at Step 2 and Step 116. This indicates that the output format is already highly stable.
> > >     2. In contrast, Table 2 demonstrates a **clear upward trend in accuracy reward**, suggesting that the model continues to improve its CoT reasoning quality over time during GRPO training.
> > >
> > > **Table 1**: Format Reward Over Training Steps
> > > Step     | Format Reward
> > > ---------|--------
> > > 0        | 1.0000
> > > 2        | 0.9974 ← slight drop
> > > 100      | 1.0000
> > > 116      | 0.9974 ← slight drop
> > > 200      | 1.0000
> > >
> > > **Table 2**: Accuracy Reward Over Training Steps
> > > Step     | Accuracy Reward
> > > ---------|--------
> > > 0        | 0.7516
> > > 50       | 0.7649
> > > 100      | 0.7891
> > > 150      | 0.8010
> > > 200      | 0.8176

---

> > > > ### Comment · Reviewer_HVvo · 2025-08-01
> > > >
> > > > Thank you for the ongoing discussion and the detailed rebuttal. Your initial responses have resolved all the concerns from my original review. I appreciate the thoroughness and have no further questions at this time.

---

> > > > > ### Author Response · Authors · 2025-08-01
> > > > >
> > > > > Dear Reviewer HVvo,
> > > > >
> > > > > We sincerely thank you for your time and thoughtful comments. We would deeply appreciate your consideration of a possible update to your rating. We greatly value your feedback and, regardless, sincerely appreciate your engagement with our work.
> > > > >
> > > > > Best regards,
> > > > >
> > > > > The Authors

---

### Official Review · Reviewer_qbY9 · 2025-06-24

**Clarity:** 3
**Significance:** 3
**Originality:** 3
**Rating:** 4
**Confidence:** 5

**Summary:**

This paper proposes UNIFIEDREWARD-THINK, a multimodal reward model that introduces long CoT reasoning to improve the accuracy and robustness of reward signals. The approach consists of three stages: initializing the model with GPT distilled data (cold start), applying rejection sampling on large-scale preference data, and using GRPO reinforcement learning to optimize reasoning capabilities. Experiments show that the method outperforms existing approaches on both visual understanding and generation tasks, and exhibits strong implicit reasoning ability even without explicitly outputting reasoning traces.

**Questions:**

1. How do you ensure the quality of the distilled cold-start data? For example, a long CoT reward dataset should guarantee both the correctness of the intermediate reasoning steps and the final scoring.

2. I'm curious why this model outperforms GPT-4o. What would happen if GPT-4o were combined with a similar long CoT reasoning approach?

**Ethical Concerns:**

["NO or VERY MINOR ethics concerns only"]

**Final Justification:**

The author has addressed my earlier questions about cumulative errors in CoT and related issues.

**Limitations:**

yes

**Quality:**

3

**Strengths And Weaknesses:**

Strengths:
1. The method is simple and easy to understand.

Weaknesses:
1. There are many R1-like papers nowadays that collect similar types of data for cold start and train with GRPO; this work lacks novelty.

2. The cold start and GRPO stages are similar to those used in most xx-R1 papers.

---

> ### Author Rebuttal · Authors · 2025-07-31
>
> Thank you for your feedback. We respectfully address your concerns as follows:
>
> ## 1. Response to Weaknesses 1&2: Regarding novelty
>
> 1. Our core innovation lies not in proposing a new technique, but in **strategically addressing a fundamental and underexplored challenge in multimodal reward model-based RL**: the unreliability of reward signals caused by direct response or shallow reasoning in existing vision reward models.
>
> 2. To the best of our knowledge, **this is the first work to introduce a unified multimodal CoT reward model**, capable of performing long CoT reward reasoning across both image and video generation and understanding tasks.
>
> 3. To address the challenge, we posit that incorporating explicit, multi-step CoT reasoning into the vision reward modeling process significantly improves the reliability and robustness of the reward signals. **However, this is not a trivial application of existing techniques**. Several unique challenges may arise in this context:
>
>    1. **Lack of multidimensional reward evaluation**: Resultant models may tend to provide oversimplified or single-facet CoT reward reasoning, which is inadequate for complex visual reward tasks. For instance, in image generation, a comprehensive evaluation should consider not only semantic consistency, but also dimensions such as authenticity and aesthetics. Neglecting any of these aspects can lead to unreliable or biased reward signals.
>    2. **Inconsistency between reasoning and final decision**: A widely observed issue in CoT-based models, where the reasoning appears implausible but the final prediction is correct. This mismatch directly undermines the reliability of the reward model, as it leads to inaccurate supervision signals during reinforcement learning.
>
> 4. To address these issues, **we introduce several key innovations in CoT reward scoring strategy design**:
>
>    1. **Multidimensional Evaluation Thinking**:
>      We design the model to perform multidimensional evaluations for each candidate (image, video, or textual response), based on the specific requirement of vision reward tasks. This enables the model to reason over diverse and task-relevant aspects, thereby enhancing the accuracy and reliability of the reward signal.
>
>    2. **Reliable Reasoning and Decision Alignment**:
>      The model is designed to explicitly assign scores to each evaluation dimension and then compute a final aggregated score across all dimensions. The candidate with the highest total score is selected as the preferred choice. This design enforces alignment between the intermediate reasoning process and the final decision, thereby mitigating a common issue in CoT-based models, i.e., the misalignment between reasoning and final prediction.
>
> 5. Based on these reward designs, we **strategically adapt and integrate post-training techniques** to elicit and reinforce CoT reasoning:
>
>    1. **Cold Start**: We first construct a distilled dataset of 5k CoT reward reasoning samples for image generation, serving as a cold start to help the model learn CoT reasoning structures.
>    2. **Rejection Sampling**: We then extend this capability to all vision tasks (image/video understanding and generation), by applying our model to large-scale unified multimodal reward datasets that contain only preference annotations (image/video/response X is better): we prompt our model to generate explicit CoT reasoning for each data. If the generated reasoning results in final prediction that aligns with the ground-truth preference, we apply rejection sampling to reinforce these correct reasoning patterns.
>    3. **GRPO**: Conversely, when the reasoning leads to incorrect predictions, we apply GRPO, which encourages the model to explore diverse reasoning paths and optimize toward outputs that better match the verified ground-truth preferences.
>
> 5. It is important to highlight that **solely relying on cold start and GRPO is insufficient** for this task due to following challenges:
>
>    - **Scalability**: Directly applying GRPO over a large scale of preference-only samples after cold start would incur high computational costs.
>    - **Low rollout quality**: After cold start, the model’s performance is still unsatisfactory as evidenced in Tables 3 and 4, making it difficult to generate correct answers during RL rollout in hard samples.
>    - **Low RL benefit on known samples**: For cases the model can already handle well after cold-start, GRPO provides limited learning gains while incurring high computation.
>
>    Therefore, we introduce **rejection sampling** as an intermediate stage to:
>    - **Filter out easy samples** that the model already gets right, reducing redundancy in GRPO, by consolidating high-quality correct reasoning patterns through these samples.
>    - **Allocate more RL effort to hard or ambiguous cases** where exploration is needed through GRPO.
>
>    We additionally include ablation results as shown in the table below. The results demonstrate that removing rejection sampling leads to a consistent drop in performance.
>
> |                         | VLReward. | GenAI-Image | GenAI-Video | VideoGen. | ShareGPTV. |
> |-------------------------|-----------|-------------|-------------|-----------|------------|
> | baseline         | 67.5      | 70.9       | 77.2        | 79.3      | 77.3       |
> | UnifiedReward-Think    | **73.8**      | **72.5**       | **82.3**        | **80.5**      | **79.6**       |
> | **w/o rejection sampling**  | 70.4  | 71.6    | 79.2    | 79.8  | 78.5   |
>
> 6. **Empirically**, our model shows substantial improvements across multiple vision reward benchmarks. Interestingly, we observe that once the model acquires long CoT reasoning ability, it can still make more accurate reward judgments even without explicitly outputting reasoning traces, suggesting **it has internalized implicit logical reasoning mechanisms**.
>
> 7. We believe our work leads to meaningful and impactful improvements in multimodal reward models and presents a valuable contribution to the community that should not be overlooked.
>
> ---
>
> ## 2. Response to Question 1: Regarding correctness of cold-start data
>
> 1. In our work, we ensure the correctness and reliability of the distilled CoT reward samples through two effective strategies:
>
>    - **Multidimensional Evaluation and Scoring Thinking**:
>      We design a multidimensional analysis and scoring CoT process to structure reasoning. Specifically, **we prompt GPT-4o to evaluate each image along several well-defined dimensions, and then aggregate the intermediate scores to arrive at a final preference decision**. This step-by-step scoring mechanism enforces alignment between reasoning and outcome, helping ensure that the final decision is grounded in interpretable and logically coherent reasoning.
>    - **Human-Aligned Ground Truth Supervision**:
>      We distill CoT reasoning samples from 10K human-preference-annotated image generation pairs, randomly sampled from four datasets: HPD, OIP, EvalMuse, and OpenAI-4o_t2i_human_preference. Each sample contains **a prompt, a pair of generated images, and a human-annotated ground-truth (GT) label** indicating the preferred image. We prompt GPT-4o to perform explicit CoT reasoning and make a comparative judgment. If **GPT-4o's final decision matches the GT label**, we retain the generated CoT reasoning and decision. This filtering process yields a final distilled dataset of approximately 5K high-quality CoT reward samples, which we use to cold-start our reward model training.
>
>    These measures jointly help ensure that the distilled reasoning traces are not only logically consistent but also aligned with human preference signals.
> 2. Furthermore, in our pipeline, the cold-start CoT data serves **primarily to teach the model the structure and format of CoT-style reasoning**, rather than to enhance reward decision quality. The generalization and strengthening of the model’s reward discrimination capability are achieved through the subsequent stages of rejection sampling and GRPO, which operate on large-scale unified multimodal vision preference data and iteratively reinforce correct reasoning trajectories.
> 3. Our experimental results in the table also show that even when using distilled CoT data from weaker models (**Qwen2.5VL-72b**) instead of GPT-4o for cold-start, the model can **still achieve comparable performance** after subsequent reinforcement through rejection sampling and GRPO.
>
> |                         | VLReward. | GenAI-Image | GenAI-Video | VideoGen. | ShareGPTV. |
> |-------------------------|-----------|-------------|-------------|-----------|------------|
> | baseline         | 67.5      | 70.9       | 77.2        | 79.3      | 77.3       |
> | UnifiedReward-Think (distilled **4o**)     | **73.8**      | 72.5       | **82.3**        | **80.5**      | 79.6       |
> | UnifiedReward-Think (distilled **Qwen2.5VL**)  | 73.2  | **73.6**    | 81.7    | 80.2  | **80.9**   |
>
> ---
> ## 3. Response to Question 2: Regarding GPT-4o's performance
> 1. While GPT-4o is a powerful general-purpose multimodal model, **it is not specifically optimized for reward modeling**, especially in terms of aligning its reward reasoning process with ground-truth human preferences across diverse image and video tasks. In contrast, **achieving reliable reward decisions requires more than general reasoning capability**. In our work, this is accomplished through our carefully designed **multidimensional evaluation and scoring reward mechanism, combined with a strategically structured three-stage training pipeline**. Together, these components ensure the generalizability and reliability of our model across both image/video understanding and generation reward tasks.
> 2. Since GPT-4o is a **closed-source model**, we are unable to directly apply our CoT-based reward modeling training pipeline to it. Nonetheless, **we believe that integrating our approach into GPT-4o would likely lead to significant improvements** in its performance on this tasks.

---

> > ### Comment · Reviewer_qbY9 · 2025-08-05
> > **Official Comment by Reviewer qbY9**
> >
> > Thank you for your detailed and well-structured response. I appreciate your identification of the issue of unreliable reasoning signals in multimodal reward modeling, and I acknowledge the effectiveness of the proposed three-stage training pipeline. I also have a few additional suggestions for your consideration:
> >
> > (1) I suggest adding a discussion on whether CoT-based reward modeling may suffer from error accumulation. Specifically, while CoT reasoning appears more interpretable than direct scoring, does it actually lead to more accurate reward estimates in practice?
> >
> > (2) It would also be helpful to include a brief analysis of how well the proposed method generalizes to out-of-distribution (OOD) tasks, and whether it can still produce reliable reward signals in such cases.
> >
> > Overall, your rebuttal has addressed most of my concerns. I will reconsider my score after further discussion with the other reviewers.

---

> > > ### Author Response · Authors · 2025-08-06
> > >
> > > Thank you for your thoughtful follow-up and recognization.
> > >
> > > 1. **Response to Suggestion 1: Regarding CoT reasoning performance in practice**
> > >     1. We appreciate the reviewer’s insightful question regarding potential error accumulation in CoT-based reward modeling and we agree it is important to assess whether it also leads to **more accurate and reliable reward estimation in practice**.
> > >
> > >     2. To address this concern, we conduct reinforcement training using **UnifiedReward-Think** on the **Flux** model via GRPO, aiming to improve its ability to generate semantically accurate images. At each training step, **UnifiedReward-Think** is employed to compute the **win rate** of each generated image against others, serving as the reward signal.
> > >     3. We evaluate the model on the widely adopted image generation benchmark **GenEval**. The following table demonstrates the effectiveness of our CoT-based reward modeling approach.
> > >     4.  As shown, compared to using UnifiedReward as the reward model, our UnifiedReward-Think provides **significantly more effective reward signals to the Flux model, leading to substantial performance improvements across multiple evaluation dimensions**. These results suggest that incorporating CoT not only enhances interpretability but also delivers **more accurate and reliable reward signals**.
> > >
> > > | Method              | Single Object | Two Object | Counting | Colors | Position | Color Attr |
> > > |---------------------|---------------|------------|----------|--------|----------|-------------|
> > > | FLUX                | 97.81         | 79.54      | 66.39    | 77.65  | 18.50    | 42.25       |
> > > | UnifiedReward       | 98.50         | 81.52      | 72.88    | 78.39  | 19.25    | 43.50       |
> > > | **UnifiedReward-Think** | **99.38**     | **83.44**  | **74.68**| **80.64**| **19.50**| **50.25**   |
> > >
> > >
> > >
> > >
> > > 2. **Response to Suggestion 2: Analysis of OOD generalization performance**
> > >     1. Our proposed three-stage training pipeline is designed to **progressively elicit and enhance the base model’s latent CoT reasoning capabilities** in multimodal reward tasks, thereby improving the accuracy, reliability, and interpretability of the reward signals. Naturally, the effectiveness of this process is influenced by the inherent capabilities of the base model itself, a stronger and more versatile base model allows for more effective reasoning and better generalization across both seen and unseen tasks.
> > >
> > >     2. Our unified multimodal CoT reward model already exhibits strong reasoning and generalization capabilities across **both image and video generation/understanding tasks**. Notably, all the benchmarks used in our study consist of **data distributions that were unseen during training**, underscoring the model’s ability to generalize to OOD scenarios. Its solid performance across a variety of multimodal reward benchmarks highlights its **rich prior knowledge and strong cross-task generalizability**.
> > >
> > >     3. To further validate our approach, we additionally evaluate our model on the **ImageReward-Bench**. As shown in the table below, **UnifiedReward-Think** delivers strong performance, outperforming all other baselines.
> > >
> > >
> > >
> > >
> > > | Method              | CLIPScore | BLIPScore | UnifiedReward | UnifiedReward-Think |
> > > |---------------------|-----------|-----------|----------------|----------------------|
> > > | ImageRewardBench | 54.82     | 57.76     | 61.81          | **65.40**            |
> > >
> > >
> > > We sincerely thank you for your time and thoughtful comments. Should you have any further suggestions or concerns, we would be more than willing to actively engage and address them promptly.

---

> > > > ### Author Response · Authors · 2025-08-09
> > > >
> > > > Dear Reviewer qbY9,
> > > >
> > > > We sincerely thank you for your time and thoughtful comments.
> > > >
> > > > We appreciate your recognition that our rebuttal has addressed most of your concerns.
> > > >
> > > > Furthermore, we have incorporated additional experiments and analyses to address the valuable supplementary suggestions you provided.
> > > >
> > > > If you believe that our response and the supplementary experimental results fully address your concerns, we would deeply appreciate your consideration of a possible update to your rating.
> > > >
> > > > Regardless, we truly value your feedback and deeply appreciate your engagement with our work.
> > > >
> > > > Best regards,
> > > >
> > > > The Authors

---

### Official Review · Reviewer_m2dN · 2025-06-26

**Clarity:** 3
**Significance:** 3
**Originality:** 3
**Rating:** 4
**Confidence:** 4

**Summary:**

This paper introduces UnifiedReward-Think, a unified multimodal reward model that enhances vision-language model alignment with human preferences by incorporating explicit Chain-of-Thought (CoT) reasoning. The key idea is to improve the reliability and robustness of reward signals in both visual understanding and generation tasks through multi-step, structured reasoning.

The proposed training framework involves three stages:
(1) Cold Start: Distill CoT reasoning traces from GPT-4o using a small image generation dataset to initialize the model with structured reasoning patterns.
(2) Rejection Sampling: Use large-scale unified preference data to fine-tune the model on correctly reasoned examples, improving generalization across tasks.
(3) GRPO-based Reinforcement Fine-Tuning: Employ Group Relative Policy Optimization on incorrectly reasoned samples, with verifiable format and accuracy rewards, to encourage exploration and correction of reasoning trajectories.

**Questions:**

1. How to guanrantee the generalizability of the resultant reward model?
2. In Section 3.3, how to know the correctness of the model-generated answers in rejection sampling?

**Ethical Concerns:**

["NO or VERY MINOR ethics concerns only"]

**Final Justification:**

Most of my concerns have been addressed, including some of the technical details, the novelty, and experimental results. I will improve my score.

**Limitations:**

yes

**Quality:**

3

**Strengths And Weaknesses:**

Strengths
1. The paper studies a timely topic, and the research problem is valuable. Aligning vision models with human preferences is an important and fundamental problem in multi-modality learning.
2. The paper writing is in general clear and easy to follow. The paper's structure is very clear. The examples provide in the paper are useful in helping understanding the proposed method.

Weaknesses
1. In Section 3.3, it is not clear how to know the correctness of the model-generated answers in rejection sampling, especially it considers both vision understanding and generation tasks.
2. In Table 2, the performance improvement of the proposed method seems to be marginal.
3. I have concerns regarding the novelty of the proposed method, since the combination of warm up + rejection sampling seems to be very standard in solving many problems.
4. The paper could be more clear about what the "reward" is intended to be used for. It seems that the reward will be used for multiple tasks. If this is the case, then the paper should emphasize why the learning can be generalized into different tasks.

---

> ### Author Rebuttal · Authors · 2025-07-31
>
> We appreciate the reviewer’s valuable feedback. We respectfully address your concerns as follows:
> ## 1. Response to Weakness 1 & Question 2: Regarding correctness of model-generated answers in rejection sampling
>   In our work, we ensure the correctness and reliability of CoT reward samples in rejection sampling through two effective strategies:
> 1. **Multidimensional CoT Scoring Mechanism**:
>     We design a multidimensional CoT analysis and scoring process to structure reasoning process. Specifically, the model is guided to **assess each candidate (image/video/response) along multiple well-defined dimensions, and then aggregate these intermediate scores to produce the final preference decision**. This step-by-step scoring mechanism enforces alignment between reasoning and outcome, helping ensure that the final decision is grounded in interpretable and logically coherent reasoning.
>
> 1. **Human Preference-Aligned Ground Truth Supervision**:
>     1. For **generation** tasks, we conduct sampling on datasets such as HPD and VideoDPO, where each sample consists of **a prompt, a pair of candidate images or videos, and human-labeled ground-truth (GT) preferences** indicating which candidate is better. During sampling, the model is prompted to perform explicit CoT reasoning to compare the two candidates. If the **final prediction of the model matches the GT label**, we consider the generated CoT reasoning and decision to be correct and apply rejection sampling to reinforce these correct reasoning patterns.
>
>     2. For **understanding** tasks, we apply a similar approach on datasets like LLaVA-Critic and ShareGPTVideo-DPO, where each sample includes **a prompt and two candidate responses, along with labeled GT preferences**. Again, if the **model’s CoT-guided final prediction aligns with the GT**, we regard the reasoning as correct and apply rejection sampling.
>
> ---
> ## 2. Response to Weakness 2: Regarding the performance improvement in Table 2
>
> 1. We would like to point out that while there is still room for improvement, our model has already achieved significant gains across most vision reward tasks, also acknowledged by Reviewer HVvo.. Specifically:
>
>    - **Image Generation Reward Task (GenAI-Bench):**
>      Our model achieves a **+1.6% gain with CoT** and **+1.0% gain without CoT** over the best baseline.
>
>    - **Video Generation Reward Task (GenAI-Bench):**
>      Our model outperforms the baseline by a significant margin of **+6.1% with CoT** and **+5.4% without CoT**, demonstrating its effectiveness in handling the complexity and temporal reasoning required for video evaluation.
>
>    - **Multimodal Understanding Task (VLRewardBench):**
>      On this challenging and widely adopted understanding benchmark, our model achieves a **+6.3% improvement with CoT** and **+5.6% without CoT** in Overall Accuracy, showing robust generalization beyond generation tasks.
>
> ---
> ## 3. Response to Weakness 3: Regarding the novelty
>
> 1. Our core innovation lies not in proposing a new technique, but in **strategically addressing a fundamental and underexplored challenge in multimodal reward model-based RL**: the unreliability of reward signals caused by direct response or shallow reasoning in existing vision reward models.
>
> 2. To the best of our knowledge, **this is the first work to introduce a unified multimodal CoT reward model**, capable of performing long CoT reward reasoning across both image and video generation and understanding tasks.
>
> 3. To address the challenge, we posit that incorporating explicit, multi-step CoT reasoning into the vision reward modeling process significantly improves the reliability and robustness of the reward signals. **However, this is not a trivial application of existing techniques**. Several unique challenges may arise in this context:
>
>    1. **Lack of multidimensional reward evaluation**: Resultant models may tend to provide oversimplified or single-facet CoT reward reasoning, which is inadequate for complex visual reward tasks. For instance, in image generation, a comprehensive evaluation should consider not only semantic consistency, but also dimensions such as authenticity and aesthetics. Neglecting any of these aspects can lead to unreliable or biased reward signals.
>    2. **Inconsistency between reasoning and final decision**: A widely observed issue in CoT-based models, where the reasoning appears implausible but the final prediction is correct. This mismatch directly undermines the reliability of the reward model, as it leads to inaccurate supervision signals during reinforcement learning.
>
> 4. To address these issues, **we introduce several key innovations in CoT reward scoring strategy design**:
>
>    1. **Multidimensional Evaluation Thinking**:
>      We design the model to perform multidimensional evaluations for each candidate (image, video, or textual response), based on the specific requirement of vision reward tasks. This enables the model to reason over diverse and task-relevant aspects, thereby enhancing the accuracy and reliability of the reward signal.
>
>    2. **Reliable Reasoning and Decision Alignment**:
>      The model is designed to explicitly assign scores to each evaluation dimension and then compute a final aggregated score across all dimensions. The candidate with the highest total score is selected as the preferred choice. This design enforces alignment between the intermediate reasoning process and the final decision, thereby mitigating a common issue in CoT-based models, i.e., the misalignment between reasoning and final prediction.
>
> 5. Based on these reward designs, we **strategically adapt and integrate post-training techniques** to elicit and reinforce CoT reasoning:
>
>    1. **Cold Start**: We first construct a distilled dataset of 5k CoT reward reasoning samples for image generation, serving as a cold start to help the model learn CoT reasoning structures.
>    2. **Rejection Sampling**: We then extend this capability to all vision tasks (image/video understanding and generation), by applying our model to large-scale unified multimodal reward datasets that contain only preference annotations (image/video/response X is better): we prompt our model to generate explicit CoT reasoning for each data. If the generated reasoning results in final prediction that aligns with the ground-truth preference, we apply rejection sampling to reinforce these correct reasoning patterns.
>    3. **GRPO**: Conversely, when the reasoning leads to incorrect predictions, we apply GRPO, which encourages the model to explore diverse reasoning paths and optimize toward outputs that better match the verified preferences.
>
> As shown in Tabs. 3 and 4, applying GRPO directly to the baseline without our CoT reward scoring strategy and the proposed three-stage training pipeline yields only limited performance gains. In contrast, our full method achieves substantially better results.
>
> 6. **Empirically**, our model shows substantial improvements across multiple vision reward benchmarks. Interestingly, we observe that once the model acquires long CoT reasoning ability, it can still make more accurate reward judgments even without explicitly outputting reasoning traces, suggesting **it has internalized implicit logical reasoning mechanisms**.
>
> 7. We believe our work leads to meaningful and impactful improvements in multimodal reward models and presents a valuable contribution to the community that should not be overlooked.
>
> ---
>
> ## 4. Response to Weakness 4 & Question 1: Regarding why the learning can be generalized to different tasks
>
> 1. In this work, our unified multimodal CoT reward model is designed to support both vision generation and understanding tasks, across both images and videos. **Although the downstream tasks may differ, the input-output structure to the reward model is unified**: in all cases, the model receives a combination of visual content (image or video) and caption or textual response, and outputs a reward judgment signal indicating the relative quality or preference between candidate outputs.
>
>    Because of this consistent structure, **the reward model learns to evaluate visual-text pairs in a task-agnostic manner**, allowing it to generalize across different modalities and task types. This design enables us to train the reward model jointly on diverse datasets and apply it effectively across vision reward tasks.
>
> 2. However, equipping reward models with CoT reasoning **using traditional training schemes like SFT poses a highly challenging problem** due to the scarcity of large-scale multimodal CoT reward data.
>
> 3. We posit that **VLMs already possess latent capabilities for multi-step reasoning**. The key is designing a scalable and effective approach to elicit and reinforce such reasoning behavior across different vision reward tasks.
>
> 4. To this end, our pipeline strategically incorporates three stages:
>    1. **Cold Start: Learning CoT Reward Format**
>    2. **Rejection Sampling: Unified CoT Reward Generalization Fine-tuning**
>    3. **GRPO: Unified CoT Reward Reinforcement Fine-tuning**
>
> 5. Through this three-stage strategy of learning, consolidation, and reinforcement, our model successfully transfers the CoT reasoning ability learned **from a small number of image generation reward tasks to all visual reward tasks**.

---

> > ### Comment · Reviewer_m2dN · 2025-08-07
> >
> > Thank you for the detailed response. I think most of my concerns have been addressed, including some of the technical details, the novelty, and experimental results. I will improve my score.

---

> > > ### Author Response · Authors · 2025-08-07
> > >
> > > Dear Reviewer m2dN,
> > >
> > > We would like to sincerely thank you for your time and insightful comments. Your thoughtful feedback is greatly appreciated, and we are truly grateful for your willingness to improve your rating. Your engagement with our work is invaluable to us.
> > >
> > > Thank you again for your constructive feedback.
> > >
> > > Best regards,
> > >
> > > The Authors

---

### Official Review · Reviewer_KVGP · 2025-07-06

**Clarity:** 3
**Significance:** 2
**Originality:** 1
**Rating:** 4
**Confidence:** 3

**Summary:**

This paper introduces UNIFIEDREWARD-THINK, a novel multimodal reward model designed to enhance the reliability and accuracy of reward signals for both visual understanding and generation tasks. The authors argue that current reward models are limited by their shallow reasoning processes and propose to integrate explicit long CoT reasoning to address this issue. The core of their methodology is a three-stage training pipeline: a "Cold Start" stage to learn the CoT format from distilled GPT-4o data, a "Rejection Sampling" stage to generalize CoT reasoning across various vision tasks, and a final "GRPO" stage to fine-tune the model on challenging cases using verifiable rewards. The authors present extensive quantitative and qualitative results demonstrating that their approach not to only improves the accuracy of reward signals with explicit CoT reasoning but also enhances the model's implicit reasoning capabilities, leading to superior performance even without the CoT.

**Questions:**

See weakness

**Ethical Concerns:**

["NO or VERY MINOR ethics concerns only"]

**Final Justification:**

I have read the new rebuttal. I think parts of my concerns have been addressed. Therefore, I raised my score.

**Limitations:**

Yes

**Quality:**

3

**Strengths And Weaknesses:**

Pros

1. The paper addresses a timely and significant problem in the field of large language models: the need for more reliable and interpretable reward models. The core idea of leveraging CoT reasoning to improve multimodal reward modeling is innovative.

2. The proposed three-stage training pipeline is logical and systematically addresses the challenges of incorporating CoT reasoning into reward models. The use of distillation for the cold start, followed by rejection sampling and reinforcement learning, is a logical and effective approach to progressively build and refine the model's reasoning abilities. And the ablation studies demonstrate the contribution of each stage to the final performance of the model.

3. The experimental results are thorough and provide strong evidence for the effectiveness of the proposed method. The authors evaluate their model on a wide range of benchmarks for both image and video understanding and generation.

Cons:

1. While the reported performance in this paper is generally strong, the work lacks clear technical novelty. The proposed method consists of three components: supervised fine-tuning (SFT), rejection sampling, and GRPO, all of which are well-established post-training techniques. When properly applied, these methods are indeed effective. However, this raises the question: What is the actual technical contribution of the paper?

2. The use of format and accuracy rewards feels somewhat superficial. Although the paper claims to promote longer chain-of-thought (CoT) reasoning, it lacks a rigorous evaluation of CoT quality. There is no explicit reward signal for distinguishing between high-quality and faulty CoTs, nor is there any post-hoc assessment of CoT correctness in the experiments. If a CoT contains factual errors or hallucinations, can it still be considered "good"? A high final accuracy alone does not necessarily indicate that the CoT itself is of high quality.

3. The results reported in Table 2 are somewhat puzzling. In several cases, the base model outperforms the trained model on all three Tau metrics, which undermines the claimed effectiveness of the proposed training approach.

---

> ### Author Rebuttal · Authors · 2025-07-31
>
> Thank you for your valuable feedback. We respectfully address your concerns as follows:
> ## 1. Response to Concern 1: Regarding novelty
> 1. Our core innovation lies not in proposing a new technique, but in **strategically addressing a fundamental and underexplored challenge in multimodal reward model-based RL**: the unreliability of reward signals caused by direct response or shallow reasoning in existing vision reward models.
>
> 2. To the best of our knowledge, **this is the first work to introduce a unified multimodal CoT reward model**, capable of performing long CoT reward reasoning across both image and video generation and understanding tasks.
>
> 3. To address the challenge, we posit that incorporating explicit, multi-step CoT reasoning into the vision reward modeling process significantly improves the reliability and robustness of the reward signals. **However, this is not a trivial application of existing techniques**. Several unique challenges may arise in this context:
>
>    1. **Lack of multidimensional reward evaluation**: Resultant models may tend to provide oversimplified or single-facet CoT reward reasoning, which is inadequate for complex visual reward tasks. For instance, in image generation, a comprehensive evaluation should consider not only semantic consistency, but also dimensions such as authenticity and aesthetics. Neglecting any of these aspects can lead to unreliable or biased reward signals.
>    2. **Inconsistency between reasoning and final decision**: A widely observed issue in CoT-based models, where the reasoning appears implausible but the final prediction is correct. This mismatch directly undermines the reliability of the reward model, as it leads to inaccurate supervision signals during reinforcement learning.
>
> 4. To address these issues, **we introduce several key innovations in CoT reward scoring strategy design**:
>
>    1. **Multidimensional Evaluation Thinking**:
>      We design the model to perform multidimensional evaluations for each candidate (image, video, or textual response), based on the specific requirement of vision reward tasks. This enables the model to reason over diverse and task-relevant aspects, thereby enhancing the accuracy and reliability of the reward signal.
>
>    2. **Reliable Reasoning and Decision Alignment**:
>      The model is designed to explicitly assign scores to each evaluation dimension and then compute a final aggregated score across all dimensions. The candidate with the highest total score is selected as the preferred choice. This design enforces alignment between the intermediate reasoning process and the final decision, thereby mitigating a common issue in CoT-based models, i.e., the misalignment between reasoning and final prediction.
>
> 5. Based on these reward designs, we **strategically adapt and integrate post-training techniques** to elicit and reinforce CoT reasoning:
>
>    1. **Cold Start**: We first construct a distilled dataset of 5k CoT reward reasoning samples for image generation, serving as a cold start to help the model learn CoT reasoning structures.
>    2. **Rejection Sampling**: We then extend this capability to all vision tasks (image/video understanding and generation), by applying our model to large-scale unified multimodal reward datasets that contain only preference annotations (image/video/response X is better): we prompt our model to generate explicit CoT reasoning for each data. If the generated reasoning results in final prediction that aligns with the ground-truth preference, we apply rejection sampling to reinforce these correct reasoning patterns.
>    3. **GRPO**: Conversely, when the reasoning leads to incorrect predictions, we apply GRPO, which encourages the model to explore diverse reasoning paths and optimize toward outputs that better match the verified preferences.
>
> As shown in Tables 3 and 4, applying GRPO directly to the baseline without our CoT reward scoring strategy and the proposed three-stage training pipeline yields only limited performance gains. In contrast, our full method achieves substantially better results.
>
> 6. **Empirically**, our model shows substantial improvements across multiple vision reward benchmarks. Interestingly, we observe that once the model acquires long CoT reasoning ability, it can still make more accurate reward judgments even without explicitly outputting reasoning traces, suggesting **it has internalized implicit logical reasoning mechanisms**.
>
> 7. We believe our work leads to meaningful and impactful improvements in enhancing the reliability of reward signals in multimodal reward models, and presents a valuable contribution to the community that should not be overlooked.
>
> ---
>
> ## 2. Response to Concern 2: Regarding CoT quality during GRPO
>
> 1. We agree that evaluating the quality of CoT reasoning is a critical aspect of ensuring the reliability of reward modeling. Indeed, **rigorously assessing reasoning quality remains a broadly acknowledged challenge across CoT-based models**. To date, there is no widely adopted and effective solution for systematically evaluating the correctness and robustness of intermediate reasoning steps during training, especially in complex multimodal tasks.
>
> 2. In our work, we mitigate this concern by **designing a multidimensional analysis and scoring CoT process**. Specifically, the model is guided to assess each candidate (image/video/response) along multiple well-defined dimensions, and then aggregate these intermediate scores to produce the final preference decision. This step-by-step structure enforces alignment between intermediate reasoning and the final outcome, helping ensure that correct decisions are derived from reliable and interpretable reasoning trajectories.
>
> 3. Thanks to this design, during GRPO training, **we treat the correctness of the final prediction as a proxy for the quality of the CoT reasoning path**. This allows us to supervise the CoT process implicitly but effectively, without requiring manual annotation of each reasoning step, which demands substantial human annotation effort.
>
> 4. In summary, **our method implicitly** **favors reasoning chains that consistently lead to reward-aligned outcomes**, which also have been discussed in Appendix A.
>
> 5. We fully acknowledge the importance of explicit post-hoc evaluation of CoT correctness, as you suggested. This is an excellent direction, and we plan to incorporate such evaluation in future iterations of our work.
>
> ---
>
> ## 3. Response to Concern 3: Regarding Tau metric
>
> 1. We would like to first clarify the interpretation of the Tau metrics reported in Table 2. In both the GenAI-Bench and VideoGen benchmarks, **the Tau metric are specifically designed to evaluate model behavior in tie cases**, where two images or videos are equally preferred.
>
> 2. In this work, our approach is explicitly focused on enhancing the model’s discriminative ability, that is, its capacity to distinguish between better and worse samples. While existing reward models are generally effective at identifying outputs with large quality differences, they often struggle to make fine-grained distinctions between two high-quality candidates. To address this gap, **we introduce a multidimensional CoT reasoning process, aimed at improving the model's sensitivity to subtle differences in quality**. As a result, we intentionally exclude tie cases from our training data, and our method is not optimized for these scenarios. This design choice is deliberate and aligns with practical objectives in vision-based reinforcement learning, where **reward models are expected to make clear and decisive judgments between competing outputs in order to effectively guide policy optimization**.
>
> 3. In contrast, baseline models were specially trained on datasets that include such tie scenarios, enabling them to perform well in such situations.
>
> 4. Although baseline models may achieve higher Tau scores, our method, both with and without CoT reasoning, **consistently demonstrates superior performance in **Diff.** metrics** where a quality difference exists between the two candidates.
>
> 5. We believe these results convincingly demonstrate the effectiveness of our proposed training approach, which is also acknowledged by Reviewer HVvo.

---

> > ### Author Response · Authors · 2025-08-07
> >
> > Dear Reviewer KVGP,
> >
> > Thank you once again for your thoughtful comments and for recognizing the importance of the problem our work addresses in the field of large language models.
> >
> > We have provided a detailed response to your concerns in the rebuttal, but we are unsure whether our reply might have been missed for any reason. We deeply value your feedback and the engaging discussion your comments have sparked.
> >
> > If you have the opportunity to revisit our response, we would be sincerely grateful for any further insights you may have. If our clarifications help to address your concerns, we would greatly appreciate your consideration in updating your score.
> >
> > Thank you again for your time and for contributing to the improvement of our work.
> >
> > Best regards,
> >
> > The Authors

---

> > > ### Comment · Reviewer_KVGP · 2025-08-08
> > >
> > > Dear Authors, I have read your rebuttal. I think your work is meaningful and well-motivated. But it still lacks enough technical contribution to be accepted by NeurIPS. I will keep my score.

---

> ### Comment · Area_Chair_fu24 · 2025-08-08
>
> Hi,
>
> Could you elaborate on your comments above? In particular, why do you think the work lacks technical contribution given the author's response, and what concrete additions do you think would improve the paper to be "above the bar"?

---

> > ### Comment · Reviewer_KVGP · 2025-08-08
> >
> > Dear AC,
> >
> > Thank you for your message and for inviting me to elaborate on my concerns.
> >
> > First, although the authors propose multidimensional evaluation, the criteria appear highly empirical. The choice and phrasing of prompts can have a large influence on results, yet there is no theoretical justification for why these dimensions are inherently valid for reward modeling, nor a discussion on how evaluation quality can be objectively measured. I did not see the authors address this limitation in depth.
> >
> > Second, the main training stages, cold start, rejection sampling, and GRPO, are well-established techniques. I did not observe clear methodological improvements or adaptations unique to this work. For example, format rewards are certainly important for training a base model, but for an instruction-tuned model with strong instruction-following capabilities (the one used in this paper), the effectiveness of this reward is questionable. The authors still directly include it, without discussion of its necessity or potential impact.
> >
> > Third, I agree that the mismatch between CoT reasoning and final prediction is a recognized challenge, and the authors’ multidimensional analysis and scoring approach could be beneficial. However, this remains highly empirical, and I consider it a significant issue that deserves much more thorough exploration, justification, and evaluation.
> >
> > For these reasons, while I find the problem important and the work well-motivated, I do not believe the technical contribution is yet above the bar for NeurIPS.

---

> > > ### Author Response · Authors · 2025-08-09
> > >
> > > Thank you for your thoughtful follow-up.
> > >
> > > 1. **Response to Concern 1: Regarding multidimensional evaluation**
> > >
> > >     1. **Validity**: We conducted relevant experiments in Tab. 3 and 4, where we compared the performance of the model when GRPO is applied solely to the final answer, without incorporating multidimensional CoT thinking in both image and video understanding/generation tasks. The results show a **noticeable improvement when multidimensional CoT thinking is included**, while the gain without CoT is minimal. We believe this experiment demonstrates both the effectiveness and necessity of multidimensional evaluation in both image and video reward tasks.
> > >
> > >     2. **Quality**: We agree that validating the quality of the CoT process is a common challenge in many CoT-related models. To address this, we ensure that the final decision is grounded in interpretable and logically coherent reasoning by incorporating **Multidimensional Evaluation Thinking** and **Reliable Reasoning and Decision Alignment**. Thanks to this design, **we treat the correctness of the final prediction as a proxy for the quality of the CoT reasoning path**. By validating the correctness of the final answer, we can implicitly and effectively assess the quality of the reasoning process.
> > >
> > > 2. **Response to Concern 2: Regarding technical contribution**
> > >
> > >     1. Our goal is not to propose a new technique, but to address a **fundamental and underexplored challenge in multimodal reward model-based RL**: the unreliability of reward signals caused by direct responses or shallow reasoning in existing vision reward models.
> > >     2. As mentioned in our rebuttal, our work **is not simply a trivial application of existing techniques**. Several unique challenges arise in this context, and we have addressed each of them individually through our approach.
> > >
> > >     2. **To the best of our knowledge, this is the first work to introduce a unified multimodal CoT reward model**, capable of performing long CoT reward reasoning across both image and video generation and understanding tasks.
> > >     3. We believe our work leads to meaningful and impactful improvements in enhancing the reliability of reward signals in multimodal reward models, and presents a valuable contribution to the community that should not be overlooked.
> > >
> > >
> > > 1. **Response to Concern 3: Regarding format reward**
> > >
> > >     1. During the GRPO stage, the inclusion of format rewards primarily **serves as a safety check to filter out occasional formatting errors**, which, although minor, can negatively impact model performance.
> > >
> > >     2. We also visualize the evolution of format rewards throughout the 200 training steps of GRPO to monitor and address any formatting inconsistencies.
> > >
> > >     3. As shown in the table, minor formatting issues do occur during training. Introducing format rewards ensures that **such errors are detected and corrected early in the training process**, promoting both consistency and reliability in the model's outputs.
> > >
> > >
> > > Table 1
> > > Step     | Format Reward
> > > ---------|--------
> > > 0        | 1.0000
> > > 2        | 0.9974 ← slight drop
> > > 100      | 1.0000
> > > 116      | 0.9974 ← slight drop
> > > 200      | 1.0000 |
> > >
> > >
> > > 4. **Response to Concern 4: Regarding mismatch between CoT reasoning and final prediction**
> > >
> > >     Thank you for your valuable comment. Could you kindly provide more detailed suggestions where you feel further exploration is necessary? We will do our best to address them thoroughly.
> > >
> > >
> > > We sincerely thank you for your time and thoughtful comments. We greatly value your feedback and, regardless, sincerely appreciate your engagement with our work.

---

### Decision · Program_Chairs · 2025-09-17

**Decision:**

Accept (poster)

**Comment:**

This paper proposes UnifiedReward-Think, a multimodal reward model that integrates chain-of-thought (CoT) reasoning into preference modeling across both vision and language tasks. The main contributions are (i) a rejection sampling strategy for generating reliable CoT supervision signals, (ii) GRPO fine-tuning to optimize over reasoning paths, and (iii) a multimodal evaluation framework. Reviewers generally agreed that the work addresses a fundamental challenge in reward modeling—the unreliability of sparse outcome-based signals—by leveraging reasoning traces to improve alignment. Strengths noted include the non-trivial novelty of combining multimodal CoT with reinforcement fine-tuning, solid theoretical motivation, and empirical gains on diverse benchmarks. Weaknesses include limited experimental breadth (a few missing baselines, not fully tested for out-of-distribution generalization) and some exposition issues in the introduction, though the rebuttal added new results, clarified distinctions from prior work, and promised revisions. After the rebuttal, most reviewers increased their confidence that the contribution is significant and technically sound. Overall, the paper offers a well-justified advance at the intersection of multimodal alignment and RLHF. I would recommend acceptance.